# Sharper Bounds for Uniformly Stable Algorithms with Stationary Mixing Processes

**Shi Fu**[1†]  **Yunwen Lei**[2∗]  **Qiong Cao**[3∗]  **Xinmei Tian**[1]    **Dacheng Tao**[3,4]

[1]University of Science and Technology of China, Hefei, China

[2]Department of Mathematics, Hong Kong Baptist University, Hong Kong, China

[3]JD Explore Academy, JD.com, Beijing, China

[4]School of Computer Science, University of Sydney, Sydney, Australia

## Abstract

Generalization analysis of learning algorithms often builds on a critical assumption that training examples are independently and identically distributed, which is often violated in practical problems such as time series prediction. In this paper, we use algorithmic stability to study the generalization performance of learning algorithms with $\psi$-mixing data, where the dependency between observations weakens over time. We show uniformly stable algorithms guarantee high-probability generalization bounds of the order $O(1/\sqrt{n})$ (within a logarithmic factor), where $n$ is the sample size. We apply our general result to specific algorithms including regularization schemes, stochastic gradient descent and localized iterative regularization, and develop excess population risk bounds for learning with $\psi$-mixing data. Our analysis builds on a novel moment bound for weakly-dependent random variables on a $\varphi$-mixing sequence and a novel error decomposition of generalization error.

## 1 Introduction

Generalization gap refers to the discrepancy between training and testing, which is a quantity of central importance in statistical learning theory (SLT) (Shalev-Shwartz & Ben-David, 2014). A popular approach to controlling the generalization gap is to bound it by the uniform convergence between training and testing errors over a function space (Bartlett & Mendelson, 2002), which leads to bounds depending on the complexity of function spaces, such as VC dimension (Vapnik, 2013), covering number (Cucker & Zhou, 2007) and Rademacher complexity (Bartlett & Mendelson, 2002). These complexity-based bounds do not exploit the property of a learning algorithm and would generally admit a square-root dependency on the dimension (Feldman, 2016), which are not favorable for large-scale problems. To incorporate the property of a learning algorithm and remove the dependency on dimension, a concept of algorithmic stability has been introduced into SLT (Bousquet & Elisseeff, 2002). Intuitively speaking, algorithmic stability measures how a small perturbation of the training dataset would affect the output model of a learning algorithm, which has close connection to several key properties such as learnability (Shalev-Shwartz et al., 2010), robustness and privacy (Bassily et al., 2020).

Recent research has witnessed an increasing interest in leveraging stability to study the generalization behavior of various algorithms, such as stochastic gradient descent (Hardt et al., 2016), structured prediction (London et al., 2016), meta learning (Maurer, 2005) and transfer learning (Kuzborskij & Lampert, 2018). Most of these discussions are based on a critical assumption that the training examples are independently and identically distributed (i.i.d.). This assumption is often violated in practical applications. For example, the i.i.d. assumption is too restrictive in time series prediction (Vidyasagar, 2013). The prices of the same stock on different days may have temporal dependence. These phenomena motivate several analyses to derive meaningful bounds for learning problems with observations drawn from a non-i.i.d. process (Yu, 1994; Vidyasagar, 2013).

A widely used relaxation of the i.i.d. assumption is to assume the observations are drawn from a mixing process (Yu, 1994; Meir, 2000; Lozano et al., 2005; Vidyasagar, 2013), where the dependency

---

[†]The work was done when Shi Fu was an intern at JD Explore Academy

[∗]Corresponding authors

between two observations is quantified by a mixing coefficient as a function of the discrepancy of the associated two indices. These mixing coefficients decay either as a polynomial function or an exponential function of the discrepancy (Vidyasagar, 2013). Several mixing processes have been introduced into the literature, including the $\beta$-mixing, $\varphi$-mixing and $\psi$-mixing processes (Yu, 1994; Meir, 2000; Lozano et al., 2005). Within this formulation, various generalization bounds have been developed to show how the dependency among observations would affect the learning process. Interestingly, these discussions imply a concept called "effective size" which plays a similar role of the sample size in the i.i.d. scenario (Yu, 1994; Kuznetsov & Mohri, 2017).

As in the i.i.d. case, most generalization analyses in the non-i.i.d. case focus on complexity-based bounds (Meir, 2000; Yu, 1994; Kuznetsov & Mohri, 2017). There are few stability analyses of learning algorithms in the non-i.i.d. cases. An exception is the work in Mohri & Rostamizadeh (2010), which, to our knowledge, gives the first systematic analysis on the stability and generalization in a non-i.i.d. case. The authors developed high-probability generalization bounds for learning with stationary $\varphi$-mixing and $\beta$-mixing sequences, which are then applied to general kernel regularization-based bounds. Due to the algorithm-specific nature, these bounds are preferable to complexity-based bounds if the associated hypothesis space has a very large complexity.

However, the stability analysis (Mohri & Rostamizadeh, 2010) only implies sub-optimal generalization bounds. Indeed, for $\beta$-uniformly stable algorithms, the high-probability bounds in Mohri & Rostamizadeh (2010) are of the order of $O(\sqrt{n}\beta + \Delta_n/\sqrt{n})$, where $n$ is the sample size and $\Delta_n$ is a term depending on the decay rate of mixing coefficients. For learning with $\lambda$-strongly convex problems, the uniform stability parameter is of the order $O(1/(n\lambda))$ (Bousquet & Elisseeff, 2002) and therefore the bounds in Mohri & Rostamizadeh (2010) become $O(1/(\sqrt{n}\lambda) + \Delta_n/\sqrt{n})$. A typical choice of $\lambda$ is $\lambda \approx n^{-\alpha}$ for $\alpha > 0$ (Shalev-Shwartz & Ben-David, 2014) and then the bounds further become $O(n^{\alpha-\frac{1}{2}} + \Delta_n/\sqrt{n})$, which cannot imply the optimal bounds $O(1/\sqrt{n})$ even if $\Delta_n = O(1)$. For learning with i.i.d. data, recent breakthroughs (Feldman & Vondrak, 2019; Bousquet et al., 2020) in stability analysis show that $\beta$-uniformly stable algorithms enjoy generalization bounds of the order $\widetilde{O}(1/\sqrt{n})^*$. This motivates a natural question: can we develop generalization bounds of the order $\widetilde{O}(1/\sqrt{n})$ for uniformly stable algorithms applied to mixing process?

This paper provides an affirmative answer to the above question. Our contributions are listed below.

1. We develop a moment bound for weakly dependent random variables defined on a $\varphi$-mixing sequence. We show our bound matches the existing moment bounds for i.i.d. random variables up to a logarithmic factor. As a byproduct, we develop a Marcinkiewicz-Zygmund inequality for a $\varphi$-mixing sequence, which may be interesting in its own right.

2. We develop high-probability bounds of order $\widetilde{O}(1/\sqrt{n})$ for uniformly stable algorithms for learning with $\psi$-mixing sequences (our results actually require assumptions on $\varphi'$-mixing coefficients which are weaker than assumptions on $\psi$-mixing coefficients). We achieve this by introducing a different decomposition of generalization errors to make sure we get weakly-dependent and mean-zero random variables, which is more challenging than the i.i.d. case. Our results recover the existing bounds within a constant factor in the i.i.d. case.

3. We apply our general bound to some specific algorithms to show the effectiveness of our results, including kernel regularization schemes, stochastic gradient descent (SGD) and iterative localization.

The paper is organized as follows. We present the related work in Section 2. We develop concentration inequalities for $\varphi$-mixing data in Section 3 and present general stability-based bounds in Section 4. We apply our general result to specific algorithms in Section 5. We conclude the paper in Section 6.

## 2 RELATED WORK

In this section, we discuss the related work. We first discuss the related work on algorithmic stability and then the related work on learning with dependent data.

**Algorithmic stability**. Algorithmic stability measures how the replacement/removal of a single (or a few) example would affect the output model, which is an important concept in SLT (Bousquet & Elisseeff, 2002). A nice property of algorithmic stability is that it only considers the behavior of the

---

*We use $\widetilde{O}$ to hide logarithmic factors

output model and therefore can imply capacity-independent generalization bounds. An important stability measure called uniform stability was introduced in an influential work (Bousquet & Elisseeff, 2002), which was used to study the generalization behavior of regularization schemes. This uniform stability was extended to the setting of randomized algorithms (Elisseeff et al., 2005), which was further used to study the generalization guarantee of SGD (Hardt et al., 2016). To better exploit the training examples for better generalization bounds, a relaxation of uniform stability called on-average stability has been introduced (Shalev-Shwartz et al., 2010). In particular, the on-average stability was shown to be equivalent to learnability (Shalev-Shwartz et al., 2010) and was used to derive data-dependent error bounds (Kuzborskij & Lampert, 2018; Lei & Ying, 2020; Zhou et al., 2021; Li et al., 2020; Nikolakakis et al., 2022). The smoothness assumption for stability analysis of SGD was removed in the papers (Lei & Ying, 2020; Bassily et al., 2020). For nonconvex problems, stability and generalization of learning algorithms that converge to global optima were studied for gradient-dominated problems (Charles & Papailiopoulos, 2018; Lei & Ying, 2021). While most discussions focus on upper bounds on the stability, recent work also develops lower bounds on the stability of SGD (Bassily et al., 2020; Amir et al., 2021). While most stability analyses imply optimal bounds in expectation, the recent study shows that uniform stability can imply almost optimal bounds with high probability (Feldman & Vondrak, 2019; Bousquet et al., 2020; Klochkov & Zhivotovskiy, 2021; Yuan & Li, 2022; Li & Liu, 2022). Algorithmic stability has found wide applications in various learning problems, including transfer learning (Kuzborskij & Lampert, 2018), meta-learning (Maurer, 2005), structured prediction (London et al., 2016), hyperparameter optimization (Bao et al., 2021), neural networks (Richards & Kuzborskij, 2021) and adversarial training (Xing et al., 2021).

**Learning with dependent data**. For learning with dependent data, one generally assumes that the data are drawn from stationary and mixing sequences with the dependence between observations diminishing appropriately over time (Doukhan, 1994; Smale & Zhou, 2009). Initially, generalization bounds were established via a uniform convergence approach based on complexity measures of function classes, such as VC dimension (Yu, 1994), covering numbers (Meir, 2000) and Rademacher complexity (Mohri & Rostamizadeh, 2008). Based on a localization idea and self-bounding loss functions, Steinwart & Christmann (2009) developed fast learning rates for regularized algorithms with geometrically $\alpha$-mixing data. Ralaivola et al. (2010) and Alquier et al. (2013) established convergence rates under the assumption of stationary and weak dependence. While most discussions focus on stationary sequence, Kuznetsov & Mohri (2017) used Rademacher complexity to study learning bounds with non-stationary $\varphi$-mixing and $\beta$-mixing sequences. The first stability analysis of learning with mixing sequences was given in the paper (Mohri & Rostamizadeh, 2010). The stability approach was also used to study online learning with dependent data (Agarwal & Duchi, 2013) and learning with graph-dependent data (Zhang et al., 2019). SGD with Markov sampling has also been recently studied (Sun et al., 2018; Wang et al., 2022).

## 3 CONCENTRATION INEQUALITIES FOR $\varphi$-MIXING SEQUENCES

We consider learning problems with a sequence of dependent observations. We assume the dependency between two observations decays with their gap. There are several concepts to quantify the dependency relationship within a stationary sequence such as $\beta$-mixing, $\varphi$-mixing and $\psi$-mixing (Mohri & Rostamizadeh, 2010; Yu, 1994). We focus on the $\varphi$-mixing and $\psi$-mixing sequences in this paper. Let $\mathbf{Z} = \{Z_t\}_{t=-\infty}^{\infty}$ be a stationary sequence of random variables. For any $i, j \in \mathbb{N}$, let $\sigma_i^j$ denote the $\sigma$-algebra generated by the random variables $Z_k, i \leq k \leq j$.

**Definition 1** ($\varphi$-Mixing Sequence). For any $k \in \mathbb{N}$, the $\varphi$-mixing coefficient of $\mathbf{Z}$ is defined as
$$\varphi(k) = \sup_{n, A \in \sigma_{n+k}^{\infty}, B \in \sigma_{-\infty}^{n}} \left| \Pr(A|B) - \Pr(A) \right|.$$

$\mathbf{Z}$ is said to be $\varphi$-mixing if $\varphi(k) \to 0$ as $k \to \infty$. It is said to be algebraically $\varphi$-mixing (with degree $r > 0$) if there exists a real number $\varphi_0 > 0$ such that $\varphi(k) \leq \varphi_0/k^r$ for all $k$, exponentially mixing (with degree $r$) if there exist real numbers $\varphi_0, \varphi_1$ such that $\varphi(k) \leq \varphi_0 \exp(-\varphi_1 k^r)$ for all $k$.

**Definition 2** ($\psi$-Mixing Sequence (Bradley, 2007)). For any $k \in \mathbb{N}$, the $\psi$-mixing coefficient of the stochastic process $\mathbf{Z}$ is defined as
$$\psi(k) = \sup_{n, A \in \sigma_{n+k}^{\infty}, B \in \sigma_{-\infty}^{n}} \left| \Pr(A \cap B)/\Pr(A)\Pr(B) - 1 \right|.$$

$\mathbf{Z}$ is said to be $\psi$-mixing if $\psi(k) \to 0$ as $k \to \infty$. It is said to be algebraically $\psi$-mixing (with degree $r > 0$) if there exists a real number $\psi_0 > 0$ such that $\psi(k) \leq \psi_0/k^r$ for all $k$, exponentially mixing (with degree $r$) if there exist real numbers $\psi_0, \psi_1$ such that $\psi(k) \leq \psi_0 \exp(-\psi_1 k^r)$ for all $k$.

We will use $\psi$-mixing to give a bound on the stability analysis version of $\varphi'$-mixing in Lemma 3. Intuitively speaking, $\varphi(k)$ and $\psi(k)$ measure the dependency of an event on those happened $k$ units of time ahead. By the definition, we know that $\psi$-mixing is stronger than the $\varphi$-mixing. Below we provide examples of $\varphi$-mixing and $\psi$-mixing sequences in Kesten & O'Brien (1976).

We first consider random variables $V_n$, $U_n$ and $S_n$ for $n \in \mathbb{Z}$ defined on the probability space $(\Omega, \mathcal{F}, P)$, which are independent of each other and have the following distributions: for all $n \in \mathbb{Z}$, $P(V_n = i) = \beta_i$, $i = 0, 1$ and $0 < \beta_0 < \beta_0 + \beta_1 = 1$; $P(U_n = k) = p_k \geq 0$, $k = 0, 1, \cdots$ and $\sum_{k=0}^{\infty} p_k = 1$; $P(S_n = j) = \gamma_j$, $j = 0, 1, 2$ and $0 < \gamma_0 < \gamma_0 + \gamma_1 < \gamma_0 + \gamma_1 + \gamma_2 = 1$.

**Example 1** (Example of $\varphi$-mixing sequence). Let $\{f_k, k \geq 1\}$ be a non-increasing sequence such that $f_1 \leq 1$, $f_k \to 0$ as $k \to \infty$ and $2\log(1 - f_{k+1}) \geq \log(1 - f_k) + \log(1 - f_{k+2})$ for $\{k : f_k < 1\}$. For any $0 < \epsilon < \frac{1}{2}$ we define $\beta_0 = \epsilon$, $\beta_1 = 1 - \epsilon$. Define $\{p_n\}$ by

$$\sum_{k=0}^{n-1} p_k = \begin{cases} (1 - f_n)(1 - f_{n+1})^{-1} & \text{if} \quad f_n < 1 \\ 0 & \text{if} \quad f_n = 1. \end{cases}$$

Note that $p_k = (1 - f_{k+1})(1 - f_{k+2})^{-1} - (1 - f_k)(1 - f_{k+1})^{-1} \geq 0$ for $\{k : f_k < 1\}$ since $2\log(1 - f_{k+1}) \geq \log(1 - f_k) + \log(1 - f_{k+2})$. Let $X_n = U_n + \frac{1}{2}V_n + \frac{1}{4}W_n$ where $W_n = V_{n-U_n}$. Then $\{X_n, n \in \mathbb{Z}\}$ is a $\varphi$-mixing sequence with $(1 - \epsilon)f_k \leq \varphi(k) \leq f_k$.

Next we provide an example of $\psi$-mixing sequence.

**Example 2** (Example of $\psi$-mixing sequence). Let $\{g_k, k \geq 1\}$ be a sequence such that $g_1 - g_2 = 1$, $g_k \to 0$ as $k \to \infty$ and $2g_{k+1} \leq g_k + g_{k+2}$, $k = 1, 2, \cdots$. For any $\epsilon \in (0, 1)$, Let $\gamma_2 = \epsilon$, $\gamma_0 = \gamma_1 = \frac{1}{2}(1 - \epsilon)$, $\beta_0 = \beta_1 = \frac{1}{2}$. Define $\{p_n\}$ by

$$p_0 = 0, \quad p_k = g_k - 2g_{k+1} + g_{k+2}, k = 1, 2, \cdots.$$

Note that $p_k \geq 0$ for $k = 0, 1, \cdots$. Let $U_n, V_n$ and $S_n$ be as before. Let $Z_n = S_n \mathbb{I}_{[S_n = 0 \text{ or } 1]} + V_{n-U_n} \mathbb{I}_{[S_n = 2]}$, where $\mathbb{I}$ is the indicator function. Finally we define $X_n = V_n + 2Z_n$. Then $\{X_n, n \in \mathbb{Z}\}$ is a $\psi$-mixing sequence with $\epsilon(1 + \epsilon)^{-1}g_k \leq \psi(k) \leq \exp\left[\epsilon(1 - \epsilon)^{-1}g_k\right] - 1$.

To develop error bounds for learning with $\varphi$-mixing sequences, we first develop concentration inequalities for $\varphi$-mixing sequences. In the following theorem to be proved in Section A, we derive a tail bound for the summation of dependent random variables in terms of the tail behavior of each individual random variable. Let $\Delta_n = 1 + 2\sum_{k=1}^{n} \varphi(k)$. The $L_p$-norm of a real-valued random variable $Z$ is denoted by $\|Z\|_p := \left(\mathbb{E}[|Z|^p]\right)^{\frac{1}{p}}, p \geq 1$.

**Theorem 1.** *Let $X_1, \ldots, X_n$ be a finite contiguous subsequence from a $\varphi$-mixing sequence. Let $Z_i$ be a function of $X_i$ with $\mathbb{E}[Z_i] = 0$ and $\Pr\{|Z_i| > \tilde{\epsilon}\} \leq 2\exp(-\tilde{\epsilon}^2/b)$. Then for any $p \geq 1$ we have*

$$\Big\|\sum_{i=1}^{n} Z_i\Big\|_p \leq (9 + \log(n))p\Delta_n\sqrt{2nb}.$$

**Remark 1.** Theorem 1 is an extension of Marcinkiewicz-Zygmund inequality for independent random variables to $\varphi$-mixing sequences. Indeed, if $Z_i$ are i.i.d., it was shown $\left\|\sum_{i=1}^{n} Z_i\right\|_p = O(p\sqrt{nb})$ (Ren & Liang, 2001). We show how the mixing behavior would affect the concentration by including $\Delta_n$ in our bound. In particular, if $Z_i$ are independent then $\Delta_n = 1$ and in this case, our result matches the Marcinkiewicz-Zygmund inequality up to a logarithmic factor. Note $\Delta_n = O(1)$ for algebraical $\varphi$-mixing with $r > 1$ and exponential $\varphi$-mixing sequences (Mohri & Rostamizadeh, 2010).

Under the assumption $\sum_{k=1}^{\infty} \varphi^{\frac{1}{2}}(k) < \infty$, it was shown $\left\|\sum_{i=1}^{n} Z_i\right\|_p \leq C_p\left(\left(\sum_{i=1}^{n} \|Z_i\|_p^p\right)^{\frac{1}{p}} + \left(\sum_{i=1}^{n} \|Z_i\|_2^2\right)^{\frac{1}{2}}\right)$ (Xuejun et al., 2010). This bound requires an assumption involving $\sum_{k=1}^{\infty} \varphi^{\frac{1}{2}}(k)$, which is larger than $\sum_{k=1}^{n} \varphi(k)$ in $\Delta_n$ since $\varphi^{\frac{1}{2}}(k) \geq \varphi(k)$. For example, if $\varphi(k) = O(k^{-1})$ then $\sum_{k=1}^{n} \varphi(k) = O(\log n)$ while $\sum_{k=1}^{n} \varphi^{\frac{1}{2}}(k) = O(\sqrt{n})$. Moreover, the bound in Xuejun et al. (2010) involves $C_p$ which is not explicitly stated. As a comparison, our bound involves all explicit constants.

**Remark 2.** Our basic idea to prove Theorem 1 is to apply a McDiarmid inequality (Lemma A.1) to a Lipschitz function defined on a $\varphi$-mixing sequence. If we define $\Phi'(X_1, \ldots, X_n) = \sum_{i=1}^{n} Z_i$, one cannot guarantee the Lipschitz continuity of $\Phi$ due to the unboundedness of $Z_i$. Our novelty is to

define $\widetilde{Z}_i = Z_i \mathbb{I}_{|Z_i| \leq \epsilon}$ where $\mathbb{I}_{[\cdot]}$ is an indicator function and $\epsilon = O(\sqrt{b \log(1/\delta)})$. The boundedness of $\widetilde{Z}_i$ implies the $(2\epsilon)$-Lipschitz continuity of $\Phi$ and therefore we can apply the McDiarmid inequality to study its decay rate of $\Phi$. Furthermore, the assumption $\Pr\{|Z_i| > \tilde{\epsilon}\} \leq 2\exp(-\tilde{\epsilon}^2/b)$ shows that $\Phi$ and $\Phi'$ are equal with a high probability. We then combine these two observations together to derive a high-probability bound for $\Phi'$, which further leads to a bound on the $L_p$-norm of $\Phi'$ by the equivalence between high-probability bound and the $L_p$-norm bound.

Based on Theorem 1, we develop a moment bound for Lipschitz functions (w.r.t. the Hamming distance) defined on mixing sequences. The following theorem is an extension of a result in the i.i.d. case (Bousquet et al., 2020) to the case with mixing sequences. This result plays a major role in developing our generalization bounds for learning with mixing sequences. The first assumption is a conditional boundedness assumption which is standard for concentration inequalities. The second assumption implies that the $g_i$ is of mean zero conditioned on any fixed $Z_{[n]\setminus[i]}$, which is stronger than $\mathbb{E}[g_i] = 0$ since the conditional expectation holds for any fixed $Z_{[n]\setminus[i]}$. The last assumption implies that $g_i$ is insensitive to the change of any single example, which implies that $g_i$ is concentrated around its expectation by McDiarmid's inequality. Our proof follows from the framework in Bousquet et al. (2020), which is given in Section B.

**Theorem 2** (Concentration Inequality for $\varphi$-Mixing Sequence). *Let $Z_1, \ldots, Z_n$ be a finite contiguous subsequence from a $\varphi$-mixing sequence. Denote $Z = \{Z_1, \ldots, Z_n\}$. Let $g_1, \ldots, g_n$ be some functions $g_i : \mathcal{Z}^n \mapsto \mathbb{R}$ such that the following holds for any $i \in [n]$*

- $\left| \mathbb{E}_{Z_{[n]\setminus[i]}}[g_i(Z)|Z_i] \right| \leq M$ *almost surely*,

- $\mathbb{E}_{Z_i}[g_i(Z)|Z_{[n]\setminus[i]}] = 0$ *a.s.*,

- $g_i$ *is $\beta$-Lipschitz w.r.t. the Hamming distance.*

*Then for any $p \geq 1$ we have ($k = \lceil \log_2 n \rceil$)*

$$\Big\| \sum_{i=1}^n g_i \Big\|_p \leq 3M\Delta_n\sqrt{2pn} + 2^k p\beta \sum_{l=0}^{k-1}(9+l)\Delta_{2^l}^2.$$

**Remark 3.** If the sequence is i.i.d., the following bound was developed (Bousquet et al., 2020)

$$\Big\| \sum_{i=1}^n g_i \Big\|_p \leq 3M\sqrt{2pn} + 12\sqrt{6}pn\beta \log_2 n. \tag{3.1}$$

Our bound recovers the existing result in the i.i.d. case. Note $\Delta_n = 1$ for any $n$ and then Theorem 2 implies $\Big\| \sum_{i=1}^n g_i \Big\|_p \leq 3M\sqrt{2pn} + np\beta(5 + \log_2\lceil n\rceil)^2$, which matches Eq. (3.1) up to a logarithmic factor. This is the first extension of the result in Bousquet et al. (2020) to a $\varphi$-mixing setting.

We follow the framework in Bousquet et al. (2020) to prove Theorem 2. The difference is to replace the Marcinkiewicz-Zygmund inequality for i.i.d. random variables by Theorem 1 for $\varphi$-mixing random variables. The basic idea is to use the representation $\sum_{i=1}^n g_i = \sum_{i=1}^n \mathbb{E}[g_i|Z_i] + \sum_{l=0}^{k-1} \sum_{i=1}^n (g_i^l - g_i^{l+1})$, where $g_i^l$ is the expectation of $g_i$ conditioned on some random variables and $k$ is an integer depending on $n$. We then use a McDiarmid inequality and the conditional boundedness of $\mathbb{E}[g_i|Z_i]$ to control $\sum_{i=1}^n \mathbb{E}[g_i|Z_i]$, and use Theorem 1 to control $\sum_{l=0}^{k-1} \sum_{i=1}^n (g_i^l - g_i^{l+1})$.

## 4 STABILITY AND GENERALIZATION

Let $\mathcal{Z} = \mathcal{X} \times \mathcal{Y}$ be a sample space, where $\mathcal{X} \subseteq \mathbb{R}^d$ is an input space and $\mathcal{Y}$ is an output space. We consider supervised learning problems where $S = \{z_1, \ldots, z_n\} = \{(x_1, y_1), \ldots, (x_n, y_n)\}$ is a contiguous subsequence from a $\psi$-mixing sequence. Based on $S$, we wish to find a model $h : \mathcal{X} \mapsto \mathcal{Y}$. We consider parametric models where the model is determined by a parameter $\mathbf{w}$ in a parameter space $\mathcal{W}$. The performance of a model $\mathbf{w}$ on a single example $z$ can be measured by a loss function $f(\mathbf{w}; z)$. The empirical risk and population risk are then defined by

$$F_S(\mathbf{w}) = \frac{1}{n}\sum_{i=1}^n f(\mathbf{w}; z_i) \quad \text{and} \quad F(\mathbf{w}) = \mathbb{E}_z[f(\mathbf{w}; z)], \tag{4.1}$$

which measure the behavior of $\mathbf{w}$ on training examples and test examples, respectively. Here the test point $z$ is assumed to be dependent on $S$ (i.e., $z$ is assumed to follow immediately after the sample $S$), which is the most realistic setting considered in Mohri & Rostamizadeh (2010). We refer to the discrepancy between training and testing as the generalization gap $F(\mathbf{w}) - F_S(\mathbf{w})$. In machine learning, we often apply an algorithm to get a model with a small training error. Meanwhile, we wish the output model also admits a small generalization gap to enjoy good generalization to test data. In this paper, we are interested in developing generalization error bounds that decay to zero as the sample size goes to infinity. Our basic tool is the algorithmic stability, which measures the sensitivity of the output up to the perturbation of a single example. Various concepts of stability have been introduced in the literature. In this paper, we focus on the uniform stability which is arguably the most widely used algorithmic stability. We use $\mathbf{w}_S$ to mean the output model if we apply an algorithm $A$ to the dataset $S$. Note we omit the dependency of the notation on $A$, which should be clear from the context. We say two sets are neighboring datasets if they differ by one example.

**Definition 3** (Uniform Stability (Bousquet & Elisseeff, 2002)). A randomized algorithm $A$ is $\epsilon$-uniformly stable if for all neighboring datasets $S, S' \in \mathcal{Z}^n$ we have $\sup_z \left[ f(\mathbf{w}_S; z) - f(\mathbf{w}_{S'}; z) \right] \leq \epsilon$.

Our stability analysis requires a different mixing coefficient defined as follows

$$\varphi'(k) = \sup_{n, A \in \sigma_{-\infty}^{n-k}, z_n \in \sigma_n^n, B \in \sigma_{n+k}^\infty} \left| \Pr(z_n | A, B) - \Pr(z_n) \right|. \tag{4.2}$$

It is clear that $\varphi'(k) \geq \varphi(k)$. The following lemma controls $\varphi'(k)$ in terms of the $\psi$-mixing coefficients. According to the following lemma, one can show that if $\psi(k) = O(k^{-r})$ then $\varphi'(k) = O(k^{-r})$. If $\psi(k) = O(\exp(-\psi_1 k^r))$ then $\varphi'(k) = O(\exp(-\psi_1 k^r))$.

**Lemma 3.** *Let* $\mathbf{Z} = \{Z_t\}_{t=-\infty}^\infty$ *be drawn from a $\psi$-mixing distribution and assume $\psi(k) < 1$. Then*

$$\varphi'(k) \leq \max \left\{ \frac{(1 + \psi(k))^2}{1 - \psi(k)} - 1, 1 - \frac{(1 - \psi(k))^2}{1 + \psi(k)} \right\}.$$

To apply Theorem 2 to learning with mixing sequences, we need to introduce a sequence of functions $g_i$ satisfying the conditions in Theorem 2 and relate them to the generalization gap. Let $z_i'$ (resp. $z_i''$) be drawn from the same distribution of $z_i$, i.e., the conditional distribution of $z_i'$ (resp. $z_i''$) given $z_1, \ldots, z_{i-1}, z_{i+1}, \ldots, z_n$ is the same as that of $z_i$ given $z_1, \ldots, z_{i-1}, z_{i+1}, \ldots, z_n$. Let $S_{i,b} = \{z_1, \ldots, z_{i-b-1}, z_i, z_{i+b+1}, \ldots, z_{n-b}\}$, i.e., we remove $2b$ points around $z_i$. For any $i \in [n]$, let $S_{i,b}^i = \{z_1, \ldots, z_{i-b-1}, z_i', z_{i+b+1}, \ldots, z_{n-b}\}$. We then define the following random variables

$$g_i = \mathbb{E}_{z_i'} \left[ \mathbb{E}_{z_i''} [f(\mathbf{w}_{S_{i,b}^i}; z_i'')] - f(\mathbf{w}_{S_{i,b}^i}; z_i) \right], \quad \forall i \in [n]. \tag{4.3}$$

The following lemma to be proved in Section C gives generalization bounds in terms of stability and $\sum_{i=1}^n g_i$. We will use $\varphi'(b)$ in Theorem 5 and all corollaries in Section 5. The underlying reason is that we need to remove $2b$ points around $z_i$ to get $S_{i,b}$ for the application of Theorem 2. An upper bound $|F(\mathbf{w}_S) - \mathbb{E}_{z_i''}[f(\mathbf{w}_{S_{i,b}}; z_i'')]|$ requires to use $\varphi'(b)$.

**Lemma 4.** *Let $S$ be drawn from a $\psi$-mixing distribution. Let $b \in \{0, \ldots, n\}$ denote the number of last points removed in $S$, i.e., $S_b = \{z_1, \ldots, z_{n-b}\}$. Let $\mathbf{w}_S$ denote the hypothesis trained on $S$. If the algorithm $A$ is $\beta$-uniformly stable and the loss function is bounded by $M > 0$, then the following inequality holds with $g_i$ defined in Eq. (4.3)*

$$\left| n(F(\mathbf{w}_S) - F_S(\mathbf{w}_S)) \right| \leq 2n(3b+1)\beta + nM(\varphi(b) + \varphi'(b)) + \left| \sum_{i=1}^n g_i \right|.$$

We can apply Theorem 2 to control the term $\sum_{i=1}^n g_i$ and derive the following generalization bounds in terms of mixing coefficients.

**Theorem 5** (General Mixing Stability Bound). *Let $\mathbf{w}_S$ denote the hypothesis returned by a $\beta$-uniformly stable algorithm trained on a sample $S$ drawn from a $\psi$-mixing stationary distribution. Let $M$ denote the uniform bound of the loss function. Then for any $b \in \{0, \ldots, n\}$ and any $\delta \in (0, 1)$, the following inequality holds with probability at least $1 - \delta$ ($k = \lceil \log_2 n \rceil$)*

$$\left| F(\mathbf{w}_S) - F_S(\mathbf{w}_S) \right| \leq 2(3b+1)\beta + M(\varphi(b) + \varphi'(b))$$

$$+ 3eM\Delta_n \sqrt{\frac{2 \log(e/\delta)}{n}} + \frac{2^{k+1} e\beta \log(e/\delta)}{n} \sum_{l=0}^{k-1} (9 + l)\Delta_{2^l}^2.$$

**Remark 4.** For $\varphi$-mixing sequences, the following high-probability bounds were developed (Mohri & Rostamizadeh, 2010)

$$\left|F(\mathbf{w}_S) - F_S(\mathbf{w}_S)\right| = O\left(\sqrt{\log(1/\delta)}\Delta_n\left(\sqrt{n}(b+1)\beta + \sqrt{n}\varphi(b) + n^{-\frac{1}{2}}\right)\right). \qquad (4.4)$$

As a comparison, our generalization bound in Theorem 5 becomes

$$\left|F(\mathbf{w}_S) - F_S(\mathbf{w}_S)\right| = O\left(\varphi'(b) + \Delta_n\sqrt{n^{-1}\log(1/\delta)} + \beta(b + \Delta_n^2\log^2 n\log(1/\delta))\right). \qquad (4.5)$$

Eq. (4.5) improves Eq. (4.4) as follows: (1) we replace $\sqrt{n}\varphi(b)$ with $\varphi'(b)$; (2) we replace $\sqrt{n}b\beta\Delta_n$ with $\beta(b + \Delta_n^2\log^2 n)$. The above two terms save a factor of $\sqrt{n}$. Meanwhile it should be also mentioned that our bounds involve $\varphi'$, while the bounds in Mohri & Rostamizadeh (2010) involve $\varphi$. We now compare these two bounds under assumptions of $\varphi'$ in three cases (results in Mohri & Rostamizadeh (2010) hold for $\varphi$-mixing coefficients and also hold for $\varphi'$-mixing coefficients).

1. In the i.i.d. case, (4.5) becomes $\left|F(\mathbf{w}_S) - F_S(\mathbf{w}_S)\right| = O\left(\sqrt{n^{-1}\log(1/\delta)} + \beta\log^2 n\log(1/\delta)\right)$, which matches existing stability-based bounds (Bousquet et al., 2020) up to a logarithmic factor. As a comparison, Eq. (4.4) becomes $\left|F(\mathbf{w}_S) - F_S(\mathbf{w}_S)\right| = O\left(\sqrt{\log(1/\delta)}(\sqrt{n}\beta + 1/\sqrt{n})\right)$.

2. We now consider algebraically mixing sequences, i.e., $\varphi'(k) \leq \varphi_0 k^{-r}$ with $r > 1$. In this case, we know $\Delta_n = O(1)$ (Mohri & Rostamizadeh, 2010). If we choose $b \asymp \beta^{-\frac{1}{r+1}}$ in Eq. (4.5), we get $\beta b \asymp b^{-r} \asymp \beta^{\frac{r}{r+1}}$ and therefore (we denote $B \asymp \widetilde{B}$ if there are absolute constants $c_1$ and $c_2$ such that $c_1 B \leq \widetilde{B} \leq c_2 B$.)

$$\left|F(\mathbf{w}_S) - F_S(\mathbf{w}_S)\right| = O\left(\beta^{\frac{r}{r+1}} + \sqrt{n^{-1}\log(1/\delta)} + \beta\log^2 n\log(1/\delta)\right). \qquad (4.6)$$

As a comparison, the optimal choice $b \asymp \beta^{-\frac{1}{r+1}}$ in Eq. (4.4) implies

$$\left|F(\mathbf{w}_S) - F_S(\mathbf{w}_S)\right| = O\left(\sqrt{\log(1/\delta)}(\sqrt{n}\beta^{\frac{r}{r+1}} + n^{-\frac{1}{2}})\right). \qquad (4.7)$$

It is clear our bound replaces $\sqrt{n}\beta^{\frac{r}{r+1}}$ in Eq. (4.7) by $\beta^{\frac{r}{r+1}} + \beta\log^2 n\log(1/\delta)$, which is smaller.

3. Finally, we consider exponential mixing sequences, i.e., $\varphi'(k) \leq \varphi_0\exp\left(-\varphi_1 k^r\right)$, which imply $\Delta_n = O(1)$. If we fix $b = \lceil\log^{\frac{1}{r}}(1/\beta)\rceil$, we know $\exp(-b^r) \leq b\beta = O(\beta\log^{\frac{1}{r}}(1/\beta))$ and

$$\left|F(\mathbf{w}_S) - F_S(\mathbf{w}_S)\right| = O\left(\sqrt{n^{-1}\log(1/\delta)} + \beta\log^2 n\log(1/\delta) + \beta\log^{\frac{1}{r}}(1/\beta)\right). \qquad (4.8)$$

As a comparison, Eq. (4.4) implies $\left|F(\mathbf{w}_S) - F_S(\mathbf{w}_S)\right| = O\left(\sqrt{\log(1/\delta)}(\sqrt{n}\beta\log^{\frac{1}{r}}(1/\beta) + \sqrt{n}\beta + 1/\sqrt{n})\right)$. It is clear that our analysis removes a factor of $\sqrt{n}$ in front of the stability parameter $\beta$.

**Remark 5.** The analysis in the mixing case is more challenging than that in the i.i.d. case. The analysis in Bousquet et al. (2020) introduces $\tilde{g}_i = \mathbb{E}_{z_i'}\mathbb{E}_{z_i}[f(\mathbf{w}_{S^i}; z) - f(\mathbf{w}_{S^i}; z_i)]$, where $z$ is independently drawn from the stationary distribution and $S^i = \{z_1, \ldots, z_{i-1}, z_i', z_{i+1}, \ldots, z_n\}$. While $\mathbb{E}_{z_i}[\tilde{g}_i] = 0$ in the i.i.d. case, we cannot guarantee $\mathbb{E}_{z_i}[\tilde{g}_i] = 0$ in the mixing case due to the dependency between $z_i$ and $z_j$ ($j \neq i$). In this way, one cannot apply Theorem 2 to $\tilde{g}_i$ in the mixing case. We use a much more complicated decomposition of the generalization error in terms of $g_i$ in Eq. (4.3), which is of mean zero in the mixing case. In this process, we introduce concepts such as $S_{i,b}, S_{i,b}^i$ to fully exploit the stability and mixing property. It should be mentioned that $S_{i,b}, S_{i,b}^i$ have been introduced in Mohri & Rostamizadeh (2010). However, the aim is different. These concepts are used in Mohri & Rostamizadeh (2010) to get bounds in *expectation* for $\Phi(S) := F(\mathbf{w}_S) - F_S(\mathbf{w}_S)$ via a lemma in Yu (1994) for $\beta$-mixing sequence, where the concentration of $\Phi(S)$ around its expectation can be directly studied via the McDiarmid inequality for $\varphi$-mixing sequence. As a comparison, our aim is to replace the sequence $\tilde{g}_i = \mathbb{E}_{z_i'}\mathbb{E}_{z_i}[f(\mathbf{w}_{S^i}; z) - f(\mathbf{w}_{S^i}; z_i)]$ (with non-zero conditional mean) by the sequence $g_i$ in Eq. (4.3) with zero conditional mean, which is then controlled by our new concentration inequality for $\varphi$-mixing sequences (Theorem 2).

## 5 APPLICATIONS

We now present applications of Theorem 5 to several algorithms, including the kernel regularization algorithm, SGD and localized iterative regularization. Let $A$ be an algorithm which outputs a model $A(S)$ after observing the dataset $S$. We are interested in the excess population risk of a model

$A(S)$ defined by $F(A(S)) - F(\mathbf{w}^*)$, which measures the performance of the output model $A(S)$ as compared to the best model $\mathbf{w}^* = \arg\min_{\mathbf{w}} F(\mathbf{w})$. An efficient approach to this aim is based on the following error decomposition (Bousquet & Bottou, 2008)

$$F(A(S)) - F(\mathbf{w}^*) = F(A(S)) - F_S(A(S)) + F_S(A(S)) - F_S(\mathbf{w}^*) + F_S(\mathbf{w}^*) - F(\mathbf{w}^*), \quad (5.1)$$

where we refer to $F(A(S)) - F_S(A(S))$ as the generalization gap and $F_S(A(S)) - F_S(\mathbf{w}^*)$ as the optimization error. The last term $F_S(\mathbf{w}^*) - F(\mathbf{w}^*)$ is easy to control since $\mathbf{w}^*$ is independent of $S$. We will apply stability analysis to control the generalization gap, and tools in optimization theory to control the optimization error. To this aim, we give some necessary definitions. The Lipschitz condition means the gradient is bounded, and the smoothness means the gradient is Lipschitz continuous. Examples of Lipschitz loss functions include the hinge loss, logistic loss and absolute loss. Examples of Lipschitz and smooth loss functions include the logistic loss and Huber loss.

**Definition 4** (Lipschitz, smoothness and convexity). Let $L, \gamma > 0$ and $\mu \geq 0$. Let $f : \mathcal{W} \times \mathcal{Z} \mapsto \mathbb{R}$.

- We say $f$ is $L$-Lipschitz continuous if $|f(\mathbf{w}; z) - f(\mathbf{w}'; z)| \leq L\|\mathbf{w} - \mathbf{w}'\|$ for any $\mathbf{w}, \mathbf{w}', z$.

- We say $f$ is $\gamma$-smooth if $\|\nabla f(\mathbf{w}; z) - \nabla f(\mathbf{w}'; z)\| \leq \gamma\|\mathbf{w} - \mathbf{w}'\|$ for any $\mathbf{w}, \mathbf{w}', z$.

- We say $f$ is $\mu$-strongly convex if $f(\mathbf{w}; z) - f(\mathbf{w}'; z) - \langle \mathbf{w} - \mathbf{w}', \nabla f(\mathbf{w}'; z)\rangle \geq \frac{\mu}{2}\|\mathbf{w} - \mathbf{w}'\|^2$ for any $\mathbf{w}, \mathbf{w}', z$. We say $f$ is convex if it is $\mu$-strongly convex with $\mu = 0$.

## 5.1 KERNEL REGULARIZATION SCHEMES

We first consider kernel regularization schemes with convex and Lipschitz loss functions. Let $K : \mathcal{X} \times \mathcal{X} \mapsto \mathbb{R}$ be a Mercer kernel (i.e., $K$ is symmetric and positive definite) and $\mathcal{W}$ be the associated reproducing kernel Hilbert space with the norm $\|\cdot\|_K$. We consider the following model

$$\mathbf{w}_{S,\lambda} = \arg\min_{\mathbf{w} \in \mathcal{W}} \{F_S(\mathbf{w}) + \lambda\|\mathbf{w}\|_K^2\}, \quad (5.2)$$

where $\lambda > 0$ is a regularization parameter to tradeoff the data-fitting term $F_S$ and the regularizer $\|\mathbf{w}\|_K^2$. The following corollary gives high-probability excess population risk bounds on kernel regularization. The proof is given in Section D.1.

**Corollary 6.** *Let $\mathbf{w}_{S,\lambda}$ denote the hypothesis returned by Eq.* (5.2) *when trained on a sample $S$ drawn from a $\psi$-mixing stationary distribution. Assume $f$ is convex, $L$-Lipschitz and bounded by $M > 0$. Then, with probability at least $1 - \delta$, the following excess risk bound holds ($k = \lceil \log_2 n \rceil$)*

$$F(\mathbf{w}_{S,\lambda}) - F(\mathbf{w}^*) = O\Big(\frac{\Delta_n \log^{\frac{1}{2}}(1/\delta)}{\sqrt{n}} + \frac{b}{n\lambda} + M\varphi'(b) + \frac{\sum_{l=0}^{k-1} l\Delta_{2^l}^2 \log(1/\delta)}{n\lambda}\Big) + \lambda\|\mathbf{w}^*\|_K^2.$$

**Remark 6.** We now instantiate the above bounds under special mixing sequences. We first consider the algebraically mixing sequence, i.e., $\varphi'(k) \leq \varphi_0 k^{-r}$ with $r > 1$. In this case, analysis similar to Remark 4 implies the following bound with an appropriate choice of $b$

$$F(\mathbf{w}_{S,\lambda}) - F(\mathbf{w}^*) = O\Big(n^{-\frac{1}{2}} \log^{\frac{1}{2}}(1/\delta) + (n\lambda)^{-\frac{r}{r+1}} + \frac{\log^2 n \log(1/\delta)}{n\lambda}\Big) + \lambda\|\mathbf{w}^*\|_K^2.$$

If $\|\mathbf{w}^*\|_K = O(1)$, then we choose $\lambda \asymp 1/\sqrt{n}$ and get

$$F(\mathbf{w}_{S,\lambda}) - F(\mathbf{w}^*) = O\Big(n^{-\frac{1}{2}} \log^{\frac{1}{2}}(1/\delta) + n^{-\frac{r}{2(r+1)}} \log^2 n \log(1/\delta)\Big).$$

We now consider the exponential mixing case, i.e., $\varphi'(k) \leq \varphi_0 \exp(-\varphi_1 k^r)$. In this case, analysis similar to Remark 4 implies

$$\big|F(\mathbf{w}_S) - F_S(\mathbf{w}_S)\big| = O\Big(\sqrt{n^{-1}\log(1/\delta)} + (n\lambda)^{-1}\log^2 n \log(1/\delta) + (n\lambda)^{-1}\log^{\frac{1}{r}}(n\lambda)\Big) + \lambda\|\mathbf{w}^*\|_K^2.$$

If $\|\mathbf{w}^*\|_K = O(1)$, then we can choose $\lambda \asymp 1/\sqrt{n}$ to derive

$$\big|F(\mathbf{w}_S) - F_S(\mathbf{w}_S)\big| = O\Big(n^{-\frac{1}{2}}\big(\sqrt{\log(1/\delta)} + \log^2 n \log(1/\delta) + \log^{\frac{1}{r}}(n)\big)\Big).$$

## 5.2 STOCHASTIC GRADIENT DESCENT

We apply our generalization bounds to SGD with convex and smooth loss functions, which has wide applications in training complex models in the big-data era due to its simplicity and efficiency.

**Definition 5** (Stochastic Gradient Descent). Let $\mathbf{w}_1 = 0 \in \mathbb{R}^d$ be an initial point and $\{\eta_t\}_t$ be a sequence of positive step sizes. SGD updates models by $\mathbf{w}_{t+1} = \mathbf{w}_t - \eta_t \nabla f(\mathbf{w}_t; z_{i_t})$, where $\nabla f(\mathbf{w}_t, z_{i_t})$ denotes a gradient of $f$ w.r.t. the first argument and $i_t$ is independently drawn from the uniform distribution over $[n] = \{1, \ldots, n\}$.

We assume the algorithm produces $\mathbf{w}_S = \frac{1}{T} \sum_{t=1}^{T} \mathbf{w}_t$, which is an average of SGD iterates. We first present the generalization error bounds. The proofs are given in Section D.2.

**Corollary 7** (Generalization bound). *Assume that the loss function $f(\cdot; z)$ is $\gamma$-smooth, convex, $L$-Lipschitz and bounded by $M > 0$ for every $z$. Suppose that we run SGD with step sizes $\eta_t \leq \min(2/\gamma, \eta)$ for $T \asymp n$ steps on a sample $S$ drawn from a $\psi$-mixing stationary distribution. Then, with probability at least $1 - \delta$ we have ($k = \lceil \log_2 n \rceil$)*

$$\left| F(\mathbf{w}_S) - F_S(\mathbf{w}_S) \right| = O\Big( \eta b \log(1/\delta) + \eta \sum_{l=0}^{k-1} l \Delta_{2^l}^2 \log^2(1/\delta) + M \varphi'(b) + \Delta_n \sqrt{\frac{\log(1/\delta)}{n}} \Big).$$

As a corollary, we develop the following excess risk bounds.

**Corollary 8** (Excess risk bound). *Assume that the loss function $f(\cdot; z)$ is $\gamma$-smooth, convex, $L$-Lipschitz and bounded by $M > 0$ for every $z$. Suppose that we run SGD with step sizes $\eta_t = \eta \asymp 1/\sqrt{T}$ for $T \asymp n$ steps on a sample $S$ drawn from a $\psi$-mixing stationary distribution. Then, with probability at least $1 - \delta$ we have ($k = \lceil \log_2 n \rceil$)*

$$F(\mathbf{w}_S) - F(\mathbf{w}^*) = O\left( n^{-\frac{1}{2}} b \log(1/\delta) + n^{-\frac{1}{2}} \sum_{l=0}^{k-1} l \Delta_{2^l}^2 \log^2(1/\delta) + M \varphi'(b) + \frac{\Delta_n \log^{\frac{1}{2}}(1/\delta) + \log^{\frac{3}{2}}(n/\delta)}{\sqrt{n}} \right).$$

## 5.3 ITERATIVE LOCALIZED ALGORITHM

We now turn to convex and non-smooth problems. In this case, SGD requires a very small step size to enjoy good stability, which however would affect the optimization process. Indeed, a tradeoff between generalization and optimization requires running SGD with $O(n^2)$ iterations, which is not computationally efficient (Lei & Ying, 2020; Bassily et al., 2020). To speed up the algorithm, we consider an iterative localization scheme (Algorithm 1 is deferred to Section D.3), which was introduced in Feldman et al. (2020). The basic idea of Algorithm 1 is to implement the optimization in epochs. At each epoch, Algorithm 1 builds an objective function with a regularizer depending on the output of the previous epoch, which is solved by SGD with $T_i$ iterations and learning rates $\{\eta_t\}$.

The following corollary gives error bounds for Algorithm 1. The proof is given in Section D.3.

**Corollary 9.** *Assume that the loss function $f(\cdot; z)$ is convex, $L$-Lipschitz and bounded by $M > 0$ for every $z$. We run Algorithm 1 on sample $S_i, i \in [m]$ drawn from a $\psi$-mixing stationary distribution. If we choose $\gamma \asymp n^{-\frac{1}{2}}$, then, with probability at least $1 - \delta$*

$$F(\mathbf{w}_m) - F(\mathbf{w}^*) = O\Big( n^{-\frac{1}{2}} \log n \Delta_n \log^{\frac{1}{2}}(n/\delta) + b n^{-\frac{1}{2}} + M \log n \varphi'(b) + n^{-\frac{1}{2}} \sum_{l=0}^{k-1} l \Delta_{2^l}^2 \log(n/\delta) \Big),$$

*where $k = \lceil \log_2 n \rceil$. Moreover, Algorithm 1 requires only $O(n \log n)$ gradient computations to achieve this generalization bound.*

## 6 CONCLUSIONS

With high probability, we develop the first stability-based generalization bounds of the order $\widetilde{O}(1/\sqrt{n})$ for learning with a mixing sequence. We apply our results to several specific algorithms such as regularization schemes, SGD and localized iterative regularization. Our analysis relies on a new moment bound for weakly-dependent random variables defined on a mixing sequence.

Our generalization bounds involve $\varphi'$-mixing coefficients, which are larger than the $\varphi$-mixing coefficients. It would be very interesting to investigate whether these $\varphi'$-mixing coefficients can be replaced by $\varphi$-mixing coefficients. We guess $\varphi'$-mixing would be more similar to $\varphi$-mixing than $\psi$-mixing since both $\varphi$- and $\varphi'$-coefficients measure the difference between a conditional probability and a probability (i.e., of the form $|\Pr(A|B) - \Pr(A)|$). As a comparison, $\psi$-mixing considers the difference between 1 and the *ratio* of probabilities (i.e. of the form $|1 - \Pr(A \cap B)/\Pr(A)\Pr(B)|$).

## ACKNOWLEDGEMENT

This work is supported by the Major Science and Technology Innovation 2030 "New Generation Artificial Intelligence" key project (No. 2021ZD0111700), NSFC No. 62222117 and National Social Science Found of China "Research on Virtual Reality Media Narrative " (Grant No.21&ZD326). The work was done when Yunwen Lei was at the School of Computer Science, University of Birmingham.

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

## A    PROOF OF THEOREM 1

In this section, we prove Theorem 1. To this aim, we introduce several lemmas. The following lemma is a McDiarmid inequality for stable functions defined on mixing sequences (Kontorovich & Ramanan, 2008).

**Lemma A.1.** *Let* $\Phi : \mathcal{Z}^n \mapsto \mathbb{R}$ *be a measurable function that is $c$-Lipschitz w.r.t. the Hamming distance for some $c > 0$, and let $Z_1, \ldots, Z_n$ be random variables distributed according to a $\varphi$-mixing distribution. Then for any $\epsilon > 0$ the following inequality holds*

$$\Pr\Big\{ \big|\Phi(Z_1, \ldots, Z_n) - \mathbb{E}[\Phi(Z_1, \ldots, Z_n)]\big| \geq \epsilon \Big\} \leq 2 \exp\Big( \frac{-2\epsilon^2}{nc^2 \Delta_n^2} \Big).$$

*Furthermore, for any $\delta \in (0, 1)$ the following inequality holds with probability at least $1 - \delta$*

$$\big|\Phi(Z_1, \ldots, Z_n) - \mathbb{E}[\Phi(Z_1, \ldots, Z_n)]\big| \leq \frac{c\Delta_n \sqrt{n \log(2/\delta)}}{\sqrt{2}}.$$

The following lemma shows the equivalence of tails and moments (Bousquet et al., 2020).

**Lemma A.2.** *Let $Y$ be a random variable. If for any $\delta \in (0,1)$, with probability at least $1 - \delta$*

$$|Y| \leq a\sqrt{\log(e/\delta)} + b\log(e/\delta),$$

*then for any $p \geq 1$ it holds that*

$$\|Y\|_p \leq 3\sqrt{p}a + 9pb.$$

*If $\|Y\|_p \leq \sqrt{p}a + pb$ for any $p \geq 1$, then for any $\delta \in (0,1)$ we have with probability at least $1 - \delta$*

$$|Y| \leq e\left(a\sqrt{\log(e/\delta)} + b\log(e/\delta)\right).$$

*Proof of Theorem 1.* Let $\epsilon > 0$ be a number to be fixed later and define

$$\widetilde{Z}_i = Z_i \mathbb{I}_{|Z_i| \leq \epsilon}, \quad \forall i \in [n],$$

where $\mathbb{I}[\cdot]$ is the indicator function (1 if the argument is true and 0 otherwise). Define $\Phi(X_1, \ldots, X_n) = \sum_{i=1}^n \widetilde{Z}_i$. First we show the Lipschitz continuity of $\Phi$ w.r.t. the Hamming distance. Suppose we change $X_1$ by $X_1'$ and keep other $X_i$. Define $Z_1', \ldots, Z_n'$ similarly to $Z_1, \ldots, Z_n$ but as functions of $X_1', X_2, \ldots, X_n$. Since $Z_j$ is a function of $X_j$ (i.e., once we know $X_j$ we know $Z_j$), we know $\widetilde{Z}_j = \widetilde{Z}_j'$ for all $j \neq 1$. Then

$$\left|\Phi(X_1, \ldots, X_n) - \Phi(X_1', X_2, \ldots, X_n)\right| = \left|\left(\widetilde{Z}_1 + \ldots + \widetilde{Z}_n\right) - \left(\widetilde{Z}_1' + \widetilde{Z}_2 + \ldots + \widetilde{Z}_n\right)\right|$$

$$= |\widetilde{Z}_1 - \widetilde{Z}_1'| \leq 2\epsilon.$$

According to Lemma A.1 with $c = 2\epsilon$ we derive the following inequality with probability at least $1 - (e - 2)\delta/e$

$$\left|\sum_{i=1}^n \widetilde{Z}_i\right| \leq \epsilon \Delta_n \sqrt{2n\log(2e/((e-2)\delta))}. \tag{A.1}$$

According to the assumption $\Pr\{|Z_i| > \tilde{\epsilon}\} \leq 2\exp(-\tilde{\epsilon}^2/b)$, we know with probability at least $1 - 2\delta/(en)$

$$|Z_i| \leq \sqrt{b\log(en/\delta)}. \tag{A.2}$$

We fix $\epsilon = \sqrt{b\log(en/\delta)}$. We now assume that Eq. (A.1) and Eq. (A.2) hold for all $i \in [n]$, which happen with probability at least $1 - \delta$. Under this event, we have $Z_i = \widetilde{Z}_i$ and (for simplicity we assume $n \geq 2/(e-2)$)

$$\left|\sum_{i=1}^n Z_i\right| = \left|\sum_{i=1}^n \widetilde{Z}_i\right| \leq \Delta_n\sqrt{2nb\log(en/\delta)\log(2e/((e-2)\delta))} \leq \sqrt{2nb}\Delta_n\log(en/\delta).$$

The following inequality then holds with probability at least $1 - \delta$

$$\left|\sum_{i=1}^n Z_i - \sqrt{2nb}\Delta_n\log(n)\right| \leq \sqrt{2nb}\Delta_n\log(e/\delta).$$

According to Lemma A.2, for any $p \geq 1$ it holds that

$$\left\|\sum_{i=1}^n Z_i - \sqrt{2nb}\Delta_n\log(n)\right\|_p \leq 9p\sqrt{2nb}\Delta_n.$$

The stated bound follows directly. The proof is completed. $\qquad\square$

## B   PROOF OF THEOREM 2

We follow the framework in Bousquet et al. (2020) to prove Theorem 2.

*Proof of Theorem 2.* Without loss of generality, we assume $n = 2^k$. Consider a sequence of partitions $\mathcal{B}_0, \ldots, \mathcal{B}_k$ with $\mathcal{B}_k = \{1, 2, \ldots, 2^k\}$. We then obtain $\mathcal{B}_l$ from $\mathcal{B}_{l+1}$ by splitting each subset in $\mathcal{B}_{l+1}$ into two equal parts. In this way, we get

$$\mathcal{B}_0 = \{\{1\}, \{2\}, \ldots, \{2^k\}\}, \ \mathcal{B}_1 = \{\{1, 2\}, \{3, 4\}, \ldots, \{2^k - 1, 2^k\}\}, \ldots, \mathcal{B}_k = \{[n]\}.$$

For each $i \in [n]$ and $l = 0, 1, \ldots, k$, denote by $B^l(i) \in \mathcal{B}_l$ the only set from $\mathcal{B}_l$ that contains $i$. In particular, $B^0(i) = \{i\}$ and $B^k(i) = [n]$.

For each $i \in [n]$ and each $l = 0, 1, \ldots, k$, introduce the random variables
$$g_i^l = g_i^l(Z_i, Z_{[n] \setminus B^l(i)}) = \mathbb{E}[g_i | Z_i, Z_{[n] \setminus B^l(i)}].$$
That is, we condition on $Z_i$ and all the variables that are not in the same set as $Z_i$ in the partition $\mathcal{B}_l$. One can check that $g_i^0 = g_i$ and $g_i^k = \mathbb{E}[g_i | Z_i]$. We can write a telescopic sum for each $i \in [n]$
$$g_i = \mathbb{E}[g_i | Z_i] + \sum_{l=0}^{k-1} (g_i^l - g_i^{l+1}).$$
It then follows from the triangle inequality that
$$\Big\| \sum_{i=1}^{n} g_i \Big\|_p \leq \Big\| \sum_{i=1}^{n} \mathbb{E}[g_i | Z_i] \Big\|_p + \sum_{l=0}^{k-1} \Big\| \sum_{i=1}^{n} (g_i^l - g_i^{l+1}) \Big\|_p. \tag{B.1}$$
Since $|\mathbb{E}[g_i | Z_i]| \leq M$, one can check that $\Phi(Z_1, \ldots, Z_n) = \sum_{i=1}^{n} \mathbb{E}[g_i | Z_i]$ is $2M$-Lipschitz w.r.t. the Hamming distance. Furthermore, we have $\mathbb{E}[\mathbb{E}[g_i | Z_i]] = 0$. Now we can apply Lemma A.1 with $c = 2M$ to derive the following inequality with at least $1 - \delta$
$$\Big| \sum_{i=1}^{n} \mathbb{E}[g_i | Z_i] \Big| \leq M \Delta_n \sqrt{2n \log(2/\delta)}.$$
It then follows from Lemma A.2 that
$$\Big\| \sum_{i=1}^{n} \mathbb{E}[g_i | Z_i] \Big\|_p \leq 3M \Delta_n \sqrt{2pn}. \tag{B.2}$$
According to the definition of $g_i^l$, one can see that
$$\mathbb{E}_{Z_{B^{l+1}(i) \setminus B^l(i)}} [g_i^l] = g_i^{l+1}.$$
Furthermore, according to our assumption we know $g_i^l$ as a function of $Z_j, j \in B^{l+1}(i) \setminus B^l(i)$ satisfies the $\beta$-Lipschitz continuity w.r.t. the Hamming distance. Therefore, one can apply Lemma A.1 with $c = \beta$ and $\Phi = g_i^l$ to derive the following inequality with (there are $2^l$ random variables)
$$\Pr\Big\{ |g_i^l - g_i^{l+1}| \geq \tilde{\epsilon} \Big\} \leq 2 \exp\Big( - \frac{2\tilde{\epsilon}^2}{\beta^2 \cdot 2^l \Delta_{2^l}^2} \Big), \tag{B.3}$$
where the probability is w.r.t. $Z_{B^{l+1}(i) \setminus B^l(i)}$.

Let us consider the sum $\sum_{i \in B^l} (g_i^l - g_i^{l+1})$ for any $B^l \in \mathcal{B}_l$. Note $Z_i' := g_i^l - g_i^{l+1}$ is a function of $Z_i, Z_{[n] \setminus B^l}$. We now condition on $Z_{[n] \setminus B^l}$ and then $Z_i'$ is a function of $Z_i$. According to Eq. (B.3), we can apply Theorem 1 with $b = 2^{l-1} \beta^2 \Delta_{2^l}^2$ to derive the following inequality
$$\mathbb{E}\Big[ \Big\| \sum_{i \in B} (g_i^l - g_i^{l+1}) \Big\|_p \Big| Z_{[n] \setminus B^l} \Big] \leq (9 + \log(|B^l|)) p \sqrt{2^l |B^l| \beta^2 \Delta_{2^l}^2 \Delta_{|B^l|}^2}$$
$$= (9 + l) p \sqrt{2^{2l} \beta^2 \Delta_{2^l}^4} = (9 + l) p 2^l \beta \Delta_{2^l}^2.$$
We now take integration w.r.t. $Z_{[n] \setminus B^l}$ and get
$$\Big\| \sum_{i \in B^l} (g_i^l - g_i^{l+1}) \Big\|_p \leq (9 + l) p 2^l \beta \Delta_{2^l}^2.$$
According to the triangle inequality, we further get
$$\Big\| \sum_{i \in [n]} (g_i^l - g_i^{l+1}) \Big\|_p \leq \sum_{B^l \in \mathcal{B}_l} \Big\| \sum_{i \in B^l} (g_i^l - g_i^{l+1}) \Big\|_p \leq 2^{k-l} \cdot (9 + l) p 2^l \beta \Delta_{2^l}^2 = (9 + l) 2^k p \beta \Delta_{2^l}^2,$$
where we have used the fact that $|\mathcal{B}_l| = 2^{k-l}$. It follows that
$$\sum_{l=0}^{k-1} \Big\| \sum_{i=1}^{n} (g_i^l - g_i^{l+1}) \Big\|_p \leq 2^k p \beta \sum_{l=0}^{k-1} (9 + l) \Delta_{2^l}^2.$$
We can plug Eq. (B.2) and the above inequality back into Eq. (B.1) to derive
$$\Big\| \sum_{i=1}^{n} g_i \Big\|_p \leq 3M \Delta_n \sqrt{2pn} + 2^k p \beta \sum_{l=0}^{k-1} (9 + l) \Delta_{2^l}^2.$$
The proof is completed. $\qquad \square$

## C PROOF OF THEOREM 5

To prove Theorem 5, we require the following lemma to control the difference between two test errors: one with the test example drawn from the mixing sequence and one with the test example drawn from the independent stationary distribution. Let $S_b = \{z_1, \ldots, z_{n-b}\}$ be the sequence by removing the last $b$ points of $S$.

**Lemma C.1** (Mohri & Rostamizadeh 2010). *Let $F(\mathbf{w}_S) = \mathrm{E}_z[f(\mathbf{w}_S; z) \mid S]$ denote the expectation in the dependent case (i.e., $z$ depends on $S$) and $\tilde{F}(\mathbf{w}_{S_b}) = \mathrm{E}_{\tilde{z}}[f(\mathbf{w}_{S_b}; \tilde{z})]$ denote the expectation where the test points are assumed independent of the training data (i.e., $\tilde{z}$ is independent of $S$). If $A$ is $\beta$-uniformly stable and $f(\mathbf{w}; z) \in [0, M]$, then the following inequality holds for any $b > 0$*

$$\left| \mathbb{E}_{\tilde{z}}[f(\mathbf{w}_{S_b}; \tilde{z})] - \mathbb{E}_z[f(\mathbf{w}_S; z) \mid S] \right| \leq b\beta + M\varphi(b).$$

*Proof of Lemma 4.* Let $z_i'$ (resp. $z_i''$) be drawn from the same distribution of $z_i$, i.e., the conditional distribution of $z_i'$ (resp. $z_i''$) given $z_1, \ldots, z_{i-1}, z_{i+1}, \ldots, z_n$ is the same as that of $z_i$ given $z_1, \ldots, z_{i-1}, z_{i+1}, \ldots, z_n$. Let $S_{i,b} = \{z_1, \ldots, z_{i-b-1}, z_i, z_{i+b+1}, \ldots, z_{n-b}\}$. For any $i \in [n]$, let $S_{i,b}^i = \{z_1, \ldots, z_{i-b-1}, z_i', z_{i+b+1}, \ldots, z_{n-b}\}$. We have the following decomposition

$$\sum_{i=1}^n \left( \mathbb{E}_{z_i''}[f(\mathbf{w}_{S_{i,b}}; z_i'')] - f(\mathbf{w}_{S_{i,b}}; z_i) \right) = \sum_{i=1}^n \left( \mathbb{E}_{z_i''}[f(\mathbf{w}_{S_{i,b}}; z_i'')] - \mathbb{E}_{z_i'}\mathbb{E}_{z_i''}[f(\mathbf{w}_{S_{i,b}^i}; z_i'')] \right)$$

$$+ \sum_{i=1}^n \mathbb{E}_{z_i'}\left[ \mathbb{E}_{z_i''}[f(\mathbf{w}_{S_{i,b}^i}; z_i'')] - f(\mathbf{w}_{S_{i,b}^i}; z_i) \right] + \sum_{i=1}^n \mathbb{E}_{z_i'}\left[ f(\mathbf{w}_{S_{i,b}^i}; z_i) - f(\mathbf{w}_{S_{i,b}}; z_i) \right].$$

According to the definition of $\beta$-uniform stability we know

$$\left| \sum_{i=1}^n \left( \mathbb{E}_{z_i''}[f(\mathbf{w}_{S_{i,b}}; z_i'')] - f(\mathbf{w}_{S_{i,b}}; z_i) \right) \right| \leq 2\beta n + \left| \sum_{i=1}^n \mathbb{E}_{z_i'}\left[ \mathbb{E}_{z_i''}[f(\mathbf{w}_{S_{i,b}^i}; z_i'')] - f(\mathbf{w}_{S_{i,b}^i}; z_i) \right] \right|.$$
(C.1)

For any $i \in [n]$, introduce

$$g_i = \mathbb{E}_{z_i'}\left[ \mathbb{E}_{z_i''}[f(\mathbf{w}_{S_{i,b}^i}; z_i'')] - f(\mathbf{w}_{S_{i,b}^i}; z_i) \right].$$

Then, we have

$$\left| \sum_{i=1}^n \left( \mathbb{E}_{z_i''}[f(\mathbf{w}_{S_{i,b}}; z_i'')] - f(\mathbf{w}_{S_{i,b}}; z_i) \right) \right| \leq 2\beta n + \left| \sum_{i=1}^n g_i \right|.$$
(C.2)

According to Lemma C.1 we know

$$\left| F(\mathbf{w}_S) - \mathbb{E}_{\tilde{z}}[f(\mathbf{w}_{S_{i,b}}; \tilde{z})] \right| \leq 3b\beta + M\varphi(b).$$

By the definition of $\varphi'$, we know

$$\left| \mathbb{E}_{z_i''}[f(\mathbf{w}_{S_{i,b}}; z_i'')] - \mathbb{E}_{\tilde{z}}[f(\mathbf{w}_{S_{i,b}}; \tilde{z})] \right| \leq M\varphi'(b).$$

Furthermore, the definition of stability implies

$$\left| f(\mathbf{w}_{S_{i,b}}; z_i) - f(\mathbf{w}_S; z_i) \right| \leq 3b\beta.$$

We combine the above three inequalities together and derive

$$\sum_{i=1}^n |F(\mathbf{w}_S) - \mathbb{E}_{z_i''}[f(\mathbf{w}_{S_{i,b}}; z_i'')]| + \sum_{i=1}^n \left| f(\mathbf{w}_{S_{i,b}}; z_i) - f(\mathbf{w}_S; z_i) \right| \leq (6b\beta + M\varphi(b) + M\varphi'(b))n.$$

Combining the above inequality and Eq. (C.1), we obtain

$$\left| n(F(\mathbf{w}_S) - F_S(\mathbf{w}_S)) \right|$$

$$\leq \left| \sum_{i=1}^n \left( \mathbb{E}_{z_i''}[f(\mathbf{w}_{S_{i,b}}; z_i'')] - f(\mathbf{w}_{S_{i,b}}; z_i) \right) \right| + \sum_{i=1}^n |F(\mathbf{w}_S) - \mathbb{E}_{z_i''}[f(\mathbf{w}_{S_{i,b}}; z_i'')]| + \sum_{i=1}^n \left| f(\mathbf{w}_{S_{i,b}}; z_i) - f(\mathbf{w}_S; z_i) \right|$$

$$\leq (6b+2)n\beta + n(M\varphi(b) + M\varphi'(b)) + \left| \sum_{i=1}^n g_i \right|.$$

The proof is completed. □

*Proof of Theorem 5.* Recall the definition of $g_i$,

$$g_i = \mathbb{E}_{z_i'}\big[\mathbb{E}_{z_i''}[f(\mathbf{w}_{S_{i,b}^i}; z_i'')] - f(\mathbf{w}_{S_{i,b}^i}; z_i)\big].$$

Since $z_i$ and $z_i''$ follow from the same distribution, we know

$$\mathbb{E}[g_i|z_{n\setminus i}] = 0.$$

One can check other assumptions in Theorem 2 also hold. Therefore, one can apply Theorem 2 to derive the following inequality with probability at least $1 - \delta$

$$\Big\|\sum_{i=1}^n g_i\Big\|_p \leq 3M\Delta_n\sqrt{2pn} + 2^{k+1}p\beta\sum_{l=0}^{k-1}(9+l)\Delta_{2^l}^2,$$

where $k = \lceil\log_2 n\rceil$. According to Lemma A.2 we further get the following inequality with probability at least $1 - \delta$

$$\sum_{i=1}^n g_i \leq e\Big(3M\Delta_n\sqrt{2n\log(e/\delta)} + 2^{k+1}\beta\sum_{l=0}^{k-1}(9+l)\Delta_{2^l}^2\log(e/\delta)\Big). \tag{C.3}$$

According to Lemma 4, we know

$$\big|n(F(\mathbf{w}_S) - F_S(\mathbf{w}_S))\big| \leq (6b+2)n\beta + n(M\varphi(b) + M\varphi'(b)) + \Big|\sum_{i=1}^n g_i\Big|.$$

We can combine the above inequality with Eq. (C.3) to derive the following inequality with probability at least $1 - \delta$

$$n\big(F(\mathbf{w}_S) - F_S(\mathbf{w}_S)\big) \leq e\Big(3M\Delta_n\sqrt{2n\log(e/\delta)} + 2^{k+1}\beta\sum_{l=0}^{k-1}(9+l)\Delta_{2^l}^2\log(e/\delta)\Big)$$
$$+ (6b+2)n\beta + n(M\varphi(b) + M\varphi'(b)).$$

The proof is completed. $\qquad\square$

# D   PROOF OF APPLICATIONS

## D.1   PROOF OF COROLLARY 6

To prove Corollary 6, we require the following lemma on the uniform stability of kernel regularization (Bousquet & Elisseeff, 2002).

**Lemma D.1** (Bousquet & Elisseeff 2002). *Let the loss function $f$ be $L$-Lipschitz and convex. Let the algorithm $A$ be defined in* (5.2). *Then $A$ is $\beta$-uniformly stable with $\beta \leq \frac{L^2}{\lambda n}$.*

*Proof of Corollary 6.* Let $A$ be the algorithm which returns $\mathbf{w}_{S,\lambda}$. By Lemma D.1, we know $A$ is $\beta$-uniformly stable, where $\beta \leq \frac{L^2}{n\lambda}$. Plugging the above inequality into Theorem 5, we derive the following inequality with probability at least $1 - \delta$

$$\big|F(\mathbf{w}_{S,\lambda}) - F_S(\mathbf{w}_{S,\lambda})\big| \leq \frac{2(3b+1)L^2}{n\lambda} + M\varphi(b) + M\varphi'(b)$$
$$+ 3eM\Delta_n\sqrt{\frac{2}{n}\log(e/\delta)} + \frac{2eL^2}{n\lambda}\sum_{l=0}^{k-1}(9+l)\Delta_{2^l}^2\log(e/\delta).$$

By Lemma A.1, we have the following inequality with probability at least $1 - \delta$

$$F_S(\mathbf{w}^*) - F(\mathbf{w}^*) = O\Big(\frac{\Delta_n\log^{\frac{1}{2}}(1/\delta)}{\sqrt{n}}\Big).$$

Furthermore, we have the following error decomposition

$F(\mathbf{w}_{S,\lambda}) - F(\mathbf{w}^*) + \lambda\|\mathbf{w}_{S,\lambda}\|_K^2$
$= F(\mathbf{w}_{S,\lambda}) - F_S(\mathbf{w}_{S,\lambda}) + F_S(\mathbf{w}_{S,\lambda}) - F_S(\mathbf{w}^*) + \lambda\|\mathbf{w}_{S,\lambda}\|_K^2 - \lambda\|\mathbf{w}^*\|_K^2 + \lambda\|\mathbf{w}^*\|_K^2 + F_S(\mathbf{w}^*) - F(\mathbf{w}^*)$
$\leq F(\mathbf{w}_{S,\lambda}) - F_S(\mathbf{w}_{S,\lambda}) + F_S(\mathbf{w}^*) - F(\mathbf{w}^*) + \lambda\|\mathbf{w}^*\|_K^2,$

where we have used the definition of $\mathbf{w}_{S,\lambda}$. We can combine the above three inequalities together to derive the stated bound. The proof is completed. $\qquad\square$

## D.2 Proof of Corollary 7 and Corollary 8

First, we prove the stability bound for convex loss minimization via SGD. Then, we apply the stability bound and Theorem 5 to the generalization bound. To develop high-probability bounds, we need to introduce a concentration inequality (Wainwright, 2019).

**Lemma D.2** (Chernoff's Bound). *Let $X_1, \ldots, X_t$ be independent random variables taking values in $\{0, 1\}$. Let $X = \sum_{j=1}^{t} X_j$ and $\mu = \mathbb{E}[X]$. Then for any $\tilde{\delta} > 0$ with probability at least $1 - \exp\left(-\mu\tilde{\delta}^2/(2 + \tilde{\delta})\right)$ we have $X \leq (1 + \tilde{\delta})\mu$. Furthermore, for any $\delta \in (0, 1)$ with probability at least $1 - \delta$ we have*

$$X \leq \mu + \log(1/\delta) + \sqrt{2\mu \log(1/\delta)}.$$

*Proof of Corollary 7.* Let $S$ and $S'$ be two samples of size $n$ differing in only a single example. Consider the gradient updates $\mathbf{w}_1, \ldots, \mathbf{w}_T$ and $\mathbf{w}'_1, \ldots, \mathbf{w}'_T$ induced by running SGD on sample $S$ and $S'$. We now suppose $S$ and $S'$ differ by the first example and apply the Lipschitz condition on $f(\cdot; z)$ to get

$$|f(\mathbf{w}_T; z) - f(\mathbf{w}'_T; z)| \leq L[\delta_T], \tag{D.1}$$

where $\delta_T = \|\mathbf{w}_T - \mathbf{w}'_T\|$. Observe that at step $t$, with probability $1 - 1/n$, the example selected by SGD is the same in both $S$ and $S'$. The convexity and $\gamma$-smoothness imply that (Hardt et al., 2016)

$$\langle \nabla f(\mathbf{v}, z) - \nabla f(\mathbf{w}, z), \mathbf{v} - \mathbf{w} \rangle \geq \frac{1}{\gamma}\|\nabla f(\mathbf{v}, z) - \nabla f(\mathbf{w}, z)\|^2 \tag{D.2}$$

If $i_t \neq 1$, by $\eta_t \leq 2/\gamma$ we know

$$\begin{aligned}
&\left\|\mathbf{w}_{t+1} - \mathbf{w}'_{t+1}\right\|^2 \\
&= \|\mathbf{w}_t - \mathbf{w}'_t\|^2 - 2\eta_t\langle \nabla f(\mathbf{w}_t, z_{i_t}) - \nabla f(\mathbf{w}'_t, z'_{i_t}), \mathbf{w}_t - \mathbf{w}'_t \rangle + \eta_t^2 \|\nabla f(\mathbf{w}_t, z_{i_t}) - \nabla f(\mathbf{w}'_t, z'_{i_t})\|^2 \\
&\leq \|\mathbf{w}_t - \mathbf{w}'_t\|^2 - \left(\frac{2\eta_t}{\gamma} - \eta_t^2\right)\|\nabla f(\mathbf{w}_t, z_{i_t}) - \nabla f(\mathbf{w}'_t, z'_{i_t})\|^2 \\
&\leq \|\mathbf{w}_t - \mathbf{w}'_t\|^2. \tag{D.3}
\end{aligned}$$

With probability $1/n$, the example selected is different, i.e. $i_t = 1$. Then, by the triangle equality and Eq. D.3,

$$\begin{aligned}
&\left\|\mathbf{w}_{t+1} - \mathbf{w}'_{t+1}\right\| \\
&= \|\mathbf{w}_t - \eta_t \nabla f(\mathbf{w}_t, z_{i_t}) - (\mathbf{w}'_t - \eta_t \nabla f(\mathbf{w}'_t, z_{i_t})\| + \eta_t\|\nabla f(\mathbf{w}'_t, z'_{i_t}) - \nabla f(\mathbf{w}'_t, z_{i_t})\| \tag{D.4} \\
&\leq \|\mathbf{w}_t - \mathbf{w}'_t\| + 2\eta_t L.
\end{aligned}$$

Combining the above two cases, we can conclude that for every $t$,

$$\left\|\mathbf{w}_{t+1} - \mathbf{w}'_{t+1}\right\| \leq \|\mathbf{w}_t - \mathbf{w}'_t\| + 2\eta_t L\mathbb{I}_{[i_t=1]}, \tag{D.5}$$

where $\mathbb{I}$ denotes the indicator function. Solving recursive inequality gives,

$$\left\|\mathbf{w}_{t+1} - \mathbf{w}'_{t+1}\right\| \leq 2L\sum_{k=1}^{t}\eta_k\mathbb{I}_{[i_k=1]} \leq 2L\eta\sum_{k=1}^{t}\mathbb{I}_{[i_k=1]}. \tag{D.6}$$

We can apply Lemma D.2 with $X_k = \mathbb{I}_{[i_k=1]}, \mu = t/n$ to get the following inequality with probability at least $1 - \delta$

$$\sum_{k=1}^{t}\mathbb{I}_{[i_k=1]} \leq t/n + \log(1/\delta) + \sqrt{2tn^{-1}\log(1/\delta)}.$$

Therefore, with probability at least $1 - \delta$, there holds

$$\left\|\mathbf{w}_{t+1} - \mathbf{w}'_{t+1}\right\| \leq 2L\eta(t/n + \log(1/\delta) + \sqrt{2tn^{-1}\log(1/\delta)}).$$

By the convexity of the norm $\|\cdot\|$, we get the following inequality with probability at least $1 - \delta$

$$\|\mathbf{w}_S - \mathbf{w}'_S\| \leq 2L\eta(T/n + \log(1/\delta) + \sqrt{2Tn^{-1}\log(1/\delta)}).$$

Plugging the inequality back into Eq. (D.1), we obtain that, with probability at least $1 - \delta$

$$|f(\mathbf{w}_S; z) - f(\mathbf{w}'_S; z)| \leq 2L^2\eta(T/n + \log(1/\delta) + \sqrt{2Tn^{-1}\log(1/\delta)}). \tag{D.7}$$

We can combine Eq. (D.7) and Theorem 5 to obtain the following inequality with probability at least $1 - \delta$

$$\left| n(F(\mathbf{w}_S) - F_S(\mathbf{w}_S)) \right| \tag{D.8}$$
$$\leq 4nL^2\eta(3b+1)(T/n + \log(2/\delta) + \sqrt{2Tn^{-1}\log(2/\delta)}) + 3eM\Delta_n\sqrt{2n\log(2e/\delta)}$$
$$+ 4nL^2\eta e\big(T/n + \log(2/\delta) + \sqrt{2Tn^{-1}\log(2/\delta)}\big)\sum_{l=0}^{k-1}(9+l)\Delta_{2^l}^2\log(\frac{2e}{\delta}) + nM(\varphi(b) + \varphi'(b)).$$

By the choice of $T \asymp n$, we can get

$$\left| n(F(\mathbf{w}_S) - F_S(\mathbf{w}_S)) \right| = O\Big(n\eta(b+1)\log(1/\delta) + nM\varphi'(b) + \Delta_n\sqrt{n\log(1/\delta)} + n\eta\log^2(1/\delta)\sum_{l=0}^{k-1}l\Delta_{2^l}^2\Big).$$

The proof is completed. $\qquad\square$

To prove excess risk bounds, we require the following high-probability bound on optimization error. Notice that optimization error analysis does not depend on the mixing property of the dataset since the randomness is taken with respect to the random indices.

**Lemma D.3** (Optimization Error (Lei & Tang, 2018)). *Assume that the loss function $f(\cdot; z)$ is convex and $L$-Lipschitz for every $z$. Suppose that we run SGD with step sizes $\eta_t = \eta \asymp \frac{1}{\sqrt{T}}$ then with probability at least $1 - \delta$ we have*

$$F_S(\mathbf{w}_S) - \inf_{\mathbf{w}} F_S(\mathbf{w}) = O(T^{-\frac{1}{2}}\log^{\frac{3}{2}}(T/\delta)).$$

*Proof of Corollary 8.* By Corollary 7, we know with probability at least $1 - \delta$ that

$$\left| F(\mathbf{w}_S) - F_S(\mathbf{w}_S) \right| = O\left(\eta b\log(1/\delta) + \eta\sum_{l=0}^{k-1}l\Delta_{2^l}^2\log^2(1/\delta) + M\varphi'(b) + \Delta_n\sqrt{\frac{\log(1/\delta)}{n}}\right).$$

By Lemma A.1, we have the following inequality with probability at least $1 - \delta$

$$F_S(\mathbf{w}^*) - F(\mathbf{w}^*) = O\Big(\frac{\Delta_n\log^{\frac{1}{2}}(1/\delta)}{\sqrt{n}}\Big).$$

Lemma D.3 shows the following inequality with probability at least $1 - \delta$

$$F_S(\mathbf{w}_S) - \inf_{\mathbf{w}} F_S(\mathbf{w}) = O(T^{-\frac{1}{2}}\log^{\frac{3}{2}}(T/\delta)).$$

We plug the above three inequalities back into Eq. (5.1), and derive the following inequality with probability at least $1 - 3\delta$

$$F(\mathbf{w}_S) - F(\mathbf{w}^*) = O\Big(n^{-\frac{1}{2}}b\log(1/\delta) + n^{-\frac{1}{2}}\sum_{l=0}^{k-1}l\Delta_{2^l}^2\log^2(1/\delta)$$
$$+ M\varphi'(b) + \frac{\Delta_n\log^{\frac{1}{2}}(1/\delta) + \log^{\frac{3}{2}}(n/\delta)}{\sqrt{n}}\Big).$$

The proof is completed. $\qquad\square$

## D.3    PROOF OF COROLLARY 9

In this section, we present the proof on the stability of the iterative localization technique. To this aim, we first present Algorithm 1.

---

**Algorithm 1:** Iterative Localized Algorithm

---

**Input:** initial point $\mathbf{w}_0 = 0$, parameter $\gamma > 0, m = \left\lceil \frac{1}{2} \log_2 n \right\rceil$

1 **for** $i = 1, 2, \ldots, m$ **do**
2    set $T_i \asymp n, \gamma_i = \frac{\gamma}{2^i}, \eta_t = \frac{\gamma_i n}{t+1}, t \in \mathbb{N}$
3    draw a sample $S_i$ of size $n$ from the mixing distribution
4    apply SGD with $T_i$ iterations and step size $\eta_t$ to minimize the following problem and get $\mathbf{w}_i$

$$\widetilde{F}_{S_i}(\mathbf{w}) := \frac{1}{n} \sum_{z \in S_i} f(\mathbf{w}; z) + \frac{1}{\gamma_i n} \|\mathbf{w} - \mathbf{w}_{i-1}\|^2 .$$

5 **end**

---

Then we move to generalization bound for Corollary 9. To this aim, We need to introduce some definitions for our proof. For any $i$, let

$$\hat{\mathbf{w}}_i = \arg \min_{\mathbf{w}} \widetilde{F}_{S_i}(\mathbf{w}), \tag{D.9}$$

where $\widetilde{F}_{S_i}$ is defined in Algorithm 1.

**Lemma D.4** (Optimization Error Bound). *Suppose that the function $\mathbf{w} \mapsto f(\mathbf{w}; z)$ is $\mu$-strongly convex (with respect to $\|\cdot\|$) and $L$-Lipschitz. Let $\{\mathbf{w}_t\}_t$ be produced by SGD on sample $S$ and step size $\eta_t = 2/(\mu(t+1))$ to minimize $F_S(\mathbf{w}) = \frac{1}{n} \sum_{i=1}^n f(\mathbf{w}; z_i)$. Set $\bar{\mathbf{w}}'_t = \left( \sum_{j=1}^t j\mathbf{w}_j \right) / \sum_{j=1}^t j$. Then, for any $\delta \in (0, 1)$, with probability at least $1 - \delta$,*

$$F_S(\bar{\mathbf{w}}'_t) - F_S(\mathbf{w}) = O(\log(1/\delta)/(t\mu)).$$

The proof of Lemma D.4 can be found in Harvey et al. (2019). According to Algorithm 1, $\mathbf{w}_i$ is the output by SGD with $\eta_t = \gamma_i n/(t+1)$ to minimize $\widetilde{F}_{S_i}(\mathbf{w})$, with the iterates weighted as Lemma D.4. The following lemma establishes the bound of Euclidean distance of $\mathbf{w}_i$ and $\hat{\mathbf{w}}_i$.

**Lemma D.5.** *Suppose that the function $\mathbf{w} \longmapsto f(\mathbf{w}; z)$ is $L$-Lipschitz and $\mu$-strongly convex. For any $\delta \in (0, 1)$, the following inequality holds with probability at least $1 - \delta$*

$$\|\hat{\mathbf{w}}_i - \mathbf{w}_i\| = O\left( \sqrt{n} \gamma_i \log^{\frac{1}{2}}(1/\delta) \right).$$

*Proof.* From Algorithm 1, we know that $\widetilde{F}_{S_i}$ is $\lambda_i := 2/(\gamma_i n)$-strongly convex. According to Lemma D.4, the following inequality holds with probability at least $1 - \delta$

$$\widetilde{F}_{S_i}(\mathbf{w}_i) - \widetilde{F}_{S_i}(\hat{\mathbf{w}}_i) = O\left(\log(1/\delta)/(T_i \lambda_i)\right) = O\left(\log(1/\delta)/(n \lambda_i)\right).$$

It then follows from the definition of $\hat{\mathbf{w}}_i$ and the strong convexity that

$$\frac{\lambda_i}{2} \|\hat{\mathbf{w}}_i - \mathbf{w}_i\|^2 \le \widetilde{F}_{S_i}(\mathbf{w}_i) - \widetilde{F}_{S_i}(\hat{\mathbf{w}}_i) = O\left(\log(1/\delta)/(n\lambda_i)\right)$$

and therefore

$$\|\hat{\mathbf{w}}_i - \mathbf{w}_i\|^2 = O\left(\log(1/\delta)/\left(n\lambda_i^2\right)\right) = O\left(n\gamma_i^2 \log(1/\delta)\right).$$

The proof is completed. $\square$

**Lemma D.6** (Bousquet & Elisseeff 2002). *Suppose the function $f : \mathcal{W} \times \mathcal{Z} \mapsto \mathbb{R}$ takes a structure $f = \ell + r$, where $\ell : \mathcal{W} \times \mathcal{Z} \mapsto \mathbb{R}$ and $r : \mathcal{W} \mapsto \mathbb{R}$. Assume for all $z$, we have $\|\nabla \ell(\mathbf{w}; z)\| \le L$. Suppose $F_S = \frac{1}{n} \sum_{i=1}^n f(\mathbf{w}; z_i)$ is $\mu$-strongly convex and define $A$ as $A(S) = \arg \min_{\mathbf{w} \in \mathcal{W}} F_S(\mathbf{w})$. Then $A$ is $\frac{4L^2}{n\mu}$-uniformly stable.*

**Lemma D.7.** *Assume for any $z, \mathbf{w} \mapsto f(\mathbf{w}; z)$ is $L$-Lipschitz. Let $\hat{\mathbf{w}}_i$ be defined in Eq. (D.9). With probability at least $1 - \delta/(2m)$ we have the following inequality uniformly for any $\mathbf{w}$*

$$\widetilde{F}_i(\hat{\mathbf{w}}_i) - \widetilde{F}_i(\mathbf{w}) = O\left( \frac{\Delta_n \log^{\frac{1}{2}}(2m/\delta)}{\sqrt{n}} + \frac{b\gamma_i}{2} + M\varphi'(b) + \frac{\gamma_i \sum_{l=0}^{k-1} l \Delta_{2^l}^2 \log(2m/\delta)}{2} \right) + \frac{2\|\mathbf{w} - \mathbf{w}_{i-1}\|^2}{\gamma_i n},$$

*where $k = \lceil \log_2 n \rceil$.*

*Proof.* For any $i$, define

$$\widetilde{F}_i(\mathbf{w}) = \mathbb{E}_z[f(\mathbf{w}; z)] + \frac{1}{\gamma_i n}\|\mathbf{w} - \mathbf{w}_{i-1}\|^2$$

and $\mathbf{w}_i^* = \arg\min_\mathbf{w} \widetilde{F}_i(\mathbf{w})$, where we assume $z$ is independently drawn from the stationary distribution of mixing sequence. Let $A_i$ be the algorithm outputting the minimizer of $\widetilde{F}_{S_i}$. We know $\widetilde{F}_{S_i}$ is $\lambda_i = 2/(\gamma_i n)$-strongly convex. Then analysis similar to Corollary 6 implies the stated inequality with probability at least $1 - \delta/(2m)$. The proof is completed. $\qquad\square$

Based on the above lemmas, we now turn to proving Corollary 9.

*Proof of Corollary 9.* Let $\hat{\mathbf{w}}_0 = \mathbf{w}^*$ and $\hat{\mathbf{w}}_i$ be defined by Eq. (D.9). We can decompose $F(\mathbf{w}_m) - F(\mathbf{w}^*)$ by

$$F(\mathbf{w}_m) - F(\mathbf{w}^*) = \sum_{i=1}^m \big(F(\hat{\mathbf{w}}_i) - F(\hat{\mathbf{w}}_{i-1})\big) + F(\mathbf{w}_m) - F(\hat{\mathbf{w}}_m). \qquad (\text{D.10})$$

Since $f$ is $L$ lipschitz, Lemma D.5 implies the following inequality with probability at least $1 - \delta/(2m)$

$$F(\mathbf{w}_m) - F(\hat{\mathbf{w}}_m) \le L\|\mathbf{w}_m - \hat{\mathbf{w}}_m\| = O(\sqrt{n}\gamma_m \log^{\frac{1}{2}}(1/\delta)). \qquad (\text{D.11})$$

Furthermore, we can apply Lemma D.7 with $\mathbf{w} = \hat{\mathbf{w}}_{i-1}$ to derive the following inequality with probability at least $1 - \delta/(2m)$

$$\widetilde{F}_i(\hat{\mathbf{w}}_i) - \widetilde{F}_i(\hat{\mathbf{w}}_{i-1}) = O\Big(\frac{\Delta_n \log^{\frac{1}{2}}(2m/\delta)}{\sqrt{n}} + \frac{b\gamma_i}{2} + M\varphi'(b) + \gamma_i \sum_{l=0}^{k-1} l\Delta_{2^l}^2 \log(2m/\delta)\Big) + \frac{2\|\hat{\mathbf{w}}_{i-1} - \mathbf{w}_{i-1}\|^2}{\gamma_i n}.$$

The following inequality then holds with probability at least $1 - \delta/2$

$$\sum_{i=1}^m \big(F(\hat{\mathbf{w}}_i) - F(\hat{\mathbf{w}}_{i-1})\big)$$
$$= \sum_{i=1}^m O\Big(\frac{\Delta_n \log^{\frac{1}{2}}(2m/\delta)}{\sqrt{n}} + b\gamma_i + M\varphi'(b) + \gamma_i \sum_{l=0}^{k-1} l\Delta_{2^l}^2 \log(2m/\delta) + \frac{\|\hat{\mathbf{w}}_{i-1} - \mathbf{w}_{i-1}\|^2}{\gamma_i n}\Big).$$

By Lemma D.5, we further get the following inequality with probability at least $1 - \delta$

$$\sum_{i=1}^m \big(F(\hat{\mathbf{w}}_i) - F(\hat{\mathbf{w}}_{i-1})\big)$$
$$= \sum_{i=1}^m O\Big(\frac{\Delta_n \log^{\frac{1}{2}}(2m/\delta)}{\sqrt{n}} + b\gamma_i + M\varphi'(b) + \gamma_i \sum_{l=0}^{k-1} l\Delta_{2^l}^2 \log(2m/\delta) + \frac{n\gamma_{i-1}^2 \log(1/\delta)}{\gamma_i n}\Big)$$
$$= \sum_{i=1}^m O\Big(\frac{\Delta_n \log^{\frac{1}{2}}(2m/\delta)}{\sqrt{n}} + b\gamma_i + M\varphi'(b) + \gamma_i \sum_{l=0}^{k-1} l\Delta_{2^l}^2 \log(2m/\delta) + \gamma_{i-1}\log(1/\delta)\Big)$$
$$= O\Big(\frac{m\Delta_n \log^{\frac{1}{2}}(2m/\delta)}{\sqrt{n}} + b\gamma + mM\varphi'(b) + \gamma \sum_{l=0}^{k-1} l\Delta_{2^l}^2 \log(2m/\delta) + \gamma\log(1/\delta)\Big),$$

where in the last two steps we have used $\gamma_i = \gamma/2^i$. We can plug the above inequality and Eq. (D.11) back into Eq. (D.10), and derive the following inequality with probability $1 - \delta$

$$F(\mathbf{w}_m) - F(\mathbf{w}^*) = O\Big(\frac{m\Delta_n \log^{\frac{1}{2}}(2m/\delta)}{\sqrt{n}} + b\gamma + mM\varphi'(b) + \gamma \sum_{l=0}^{k-1} l\Delta_{2^l}^2 \log(2m/\delta)$$
$$+ \sqrt{n}\gamma_m \log^{\frac{1}{2}}(1/\delta)\Big).$$

This gives the stated bound. The proof is completed. $\qquad\square$

# E   PROOF OF LEMMA 3

*Proof of Lemma 3.* For simplicity, we only consider discrete random variables. Let $A \in \sigma_{-\infty}^{n-k}, B \in \sigma_{n+k}^{\infty}$ and $Z_n, z \in \sigma_n^n$. According to the definition of mixing sequence, we know

$$1 - \psi(k) \leq \frac{\Pr(A|B)}{\Pr(A)} \leq \psi(k) + 1. \tag{E.1}$$

By Eq. (E.1) we know

$$
\begin{aligned}
\Pr(Z_n|A, B) &= \frac{\Pr(Z_n, A, B)}{\Pr(A, B)} = \frac{\Pr(Z_n, A, B)}{\sum_z \Pr(z, A, B)} \\
&= \frac{\Pr(A)\Pr(Z_n|A)\Pr(B|A, Z_n)}{\sum_z \Pr(A)\Pr(z|A)\Pr(B|A, z)} = \frac{\Pr(Z_n|A)\Pr(B|A, Z_n)}{\sum_z \Pr(z|A)\Pr(B|A, z)} \\
&\geq \frac{\Pr(Z_n|A)\Pr(B|A, Z_n)}{\sum_z \Pr(z|A)\big(\Pr(B) + \psi(k)\Pr(B)\big)} = \frac{\Pr(Z_n|A)\Pr(B|A, Z_n)}{\Pr(B) + \psi(k)\Pr(B)} \\
&\geq \frac{\Pr(Z_n|A)\Pr(B)(1 - \psi(k))}{\Pr(B) + \psi(k)\Pr(B)} = \frac{\Pr(Z_n|A)(1 - \psi(k))}{1 + \psi(k)} \\
&\geq \frac{\Pr(Z_n)(1 - \psi(k))^2}{1 + \psi(k)}.
\end{aligned}
$$

It then follows that

$$\Pr(Z_n|A, B) - \Pr(Z_n) \geq \frac{\Pr(Z_n)(1 - \psi(k))^2}{1 + \psi(k)} - \Pr(Z_n) = \Pr(Z_n)\Big(\frac{(1 - \psi(k))^2}{1 + \psi(k)} - 1\Big).$$

In a similar way, one can show

$$\Pr(Z_n|A, B) - \Pr(Z_n) \leq \Pr(Z_n)\Big(\frac{(1 + \psi(k))^2}{1 - \psi(k)} - 1\Big).$$

The proof is completed by combining the above two inequalities together. $\qquad\square$

