# OpenReview forum: "Sharper Bounds for Uniformly Stable Algorithms with Stationary Mixing Process"
_ICLR.cc/2023/Conference — ICLR 2023 poster_

### Official Review · Reviewer_2VCS · 2022-10-24

**Confidence:** 3
**Correctness:** 3
**Technical Novelty And Significance:** 2
**Empirical Novelty And Significance:** Not applicable
**Recommendation:** 6

**Clarity, Quality, Novelty And Reproducibility:**

The paper is written very well which makes it easy to understand the main ideas behind the analysis. I believe the novelty of the results is limited as the main technical tools used in the analysis (Theorems 1 and 2) seem to follow quite readily from existing results. This is explained in the weaknesses section in more detail. Nevertheless, the quality of the results is decent in my view as it improves upon the existing result of (Mohri & Rostamizadeh, 2010) for non iid data generated from a mixing sequence.

**Strength And Weaknesses:**

Pros: The paper has the following strengths.

• Very well written paper with a clear exposition that explains the problem setup, main contributions, related work, and the technical ideas cleanly.

• The high probability generalization error bounds for learning algorithms trained on non iid data generated from a mixing sequence is a solid addition to the literature, and novel to my knowledge. It improves the existing bound of (Mohri & Rostamizadeh, 2010) by a $\sqrt{n}$ factor.

Cons: The paper has the following weaknesses.

• Theorems 1 and 2 are the main technical tools used for deriving the main results. However, the proof of Theorem 2 essentially follows the telescoping argument of (Bousquet et al. 2020) with little changes. And the proof of Theorem 1 follows readily from Lemma A.1 (as mentioned above), so there does not seem to be much technical novelty in the analysis.

• I think it would be helpful for the reader if some discussion on the proof steps can be provided within the main text, especially for Theorems 1 and 2 as these are the main technical tools needed for deriving the main results.  At the moment, no such discussion exists regarding the proof steps.

Further comments:

•  In Remark 4 in the fourth line, should it be “we cannot guarantee $\mathbb{E}[\tilde{g}_i] = 0$…”?

• Theorem 5 is for loss functions which are uniformly bounded by M. It would be helpful to clarify in the text how the conditions required within the corollaries in Section 5 handle this boundedness condition. For example, in the proof of Corollary 6, it is not clear to me from the proof where M disappears?

• In the statement of Corollary 6, I think f should be stated to be convex?



**Summary Of The Paper:**

The paper studies the generalization performance of learning algorithms in a non iid setting, via the lens of algorithmic stability. The assumption made on the data is that it is generated by a mixing process ($\phi$ mixing or $\psi$ mixing) which formally quantifies how the level of dependency between two observations decays as a function of the gap between their indices. The paper derives moment bounds for weakly dependent random variables, which leads to a generalization of the Marcinkiewicz-Zygmund inequality (for iid random variables) to the non iid setting for mixing sequences. Then following the ideas in (Bousquet et al., 2020), they derive high probability bounds on the generalization error for uniformly stable learning algorithms trained on a sample drawn from a mixing process, with bounded loss functions. Applications of this bound are shown for several settings such as kernel regularization, stochastic gradient descent and iterative localized optimization algorithms.

**Summary Of The Review:**

The paper is very well written. The problem setup, related work, and the setting considered is explained clearly, and the exposition of the technical results are also clean. I think this is a decent result and is a natural extension of the recent advancements made in the context of high probability bounds for the iid setting to the non iid setup. The bounds improve that of (Mohri & Rostamizadeh, 2010) by a factor of $\sqrt{n}$ up to log factors, and this is made possible by heavily relying on the proof steps of (Bosquet et al. 2020), which leads to Theorem 2. The Marcinkiewicz-Zygmund type inequality developed in Theorem 1 is a useful tool although it seems to follow quite readily from the Mcdiarmid inequality in Lemma A.1 (Kontorovich and Ramanan, 2008).

---

> ### Author Response · Authors · 2022-11-16
> **Response to Reviewer 2VCS**
>
> Thank you for your time and efforts. Below please find the responses to some specific comments.
>
> **Q1: The proof of Theorem 2 essentially follows the telescoping argument of (Bousquet et al. 2020) with little changes. The proof of Theorem 1 follows readily from Lemma A.1 (as mentioned above), so there does not seem to be much technical novelty in the analysis.**
>
> **A**: Thank you for the comments. Our novelty of proving Theorem 1 is to introduce an indicator function to get a Lipschitz function, which is required for the application of the McDiarmid inequality. We have added Remark 2 in the revision to clarify the novelty. We follow the framework in Bousquet et al. (2020) to prove Theorem 2. The difference is to replace the Marcinkiewicz-Zygmund inequality for i.i.d. random variables by Theorem 1 for $\varphi$-mixing random variables. Our main technical difference from Bousquet et al. (2020) is to use a new decomposition of the generalization error into $g_i$ in the proof of Theorem 5. To address the dependency among training examples, we remove $2b$ points around $z_i$ in the construction of $g_i$, which is not required in the i.i.d. case. We have added some discussions in Remark 5.
>
> **Q2: It would be helpful for the reader if some discussion on the proof steps can be provided within the main text, especially for Theorems 1 and 2.**
>
> **A**:  Thank you for the constructive comment. We have added discussions on the proof of Thm 1 in Remark 2 as follows:
>
> Our basic idea to prove Thm 1 is to apply a McDiarmid inequality to a Lipschitz function defined on a $\varphi$-mixing sequence. If we define $\Phi'(X_1,\ldots,X_n)=\sum_{i=1}^{n}Z_i$, one cannot guarantee the Lipschitz continuity of $\Phi$ due to the unboundedness of $Z_i$. Our novelty is to define $\widetilde{Z}\_i=Z_iI_{|Z_i|\leq \epsilon}$ where $I_{[\cdot]}$ is an indicator function and $\epsilon=O(\sqrt{b\log(1/\delta)})$. The boundedness of $\widetilde{Z}\_i$ implies the $(2\epsilon)$-Lipschitz continuity of $\Phi$ and therefore we can apply the McDiarmid inequality to study its decay rate of $\Phi$. Furthermore, the assumption $\mathrm{Pr}\\{|Z_i|>\tilde{\epsilon}\\}\leq2\exp(-\tilde{\epsilon}\^2/b)$ shows that $\Phi$ and $\Phi'$ are equal with a high probability. We then combine these two observations together to derive a high-probability bound for $\Phi'$, which further leads to a bound on the $L_p$-norm of $\Phi'$ by the equivalence between high-probability bound and the $L_p$-norm bound.
>
> We have added discussions on the proof of Thm 2 in Remark 3 as follows:
>
> We follow the framework in Bousquet et al. (2020) to prove Thm 2. The difference is to replace the Marcinkiewicz-Zygmund inequality for i.i.d. random variables by Thm 1 for $\varphi$-mixing random variables. The basic idea is to use the representation
>
> $$
>   \sum_{i=1}^{n}g_i=\sum_{i=1}^{n}\mathbb{E}[g_i|Z_i]+\sum_{l=0}^{k-1}\sum_{i=1}^{n}\big(g_i^l-g_i^{l+1}\big),
> $$
>
>   where $g_i^l$ is the expectation of $g_i$ conditioned on some random variables and $k$ is an integer depending on $n$. We then use a McDiarmid inequality and the conditional boundedness of $\mathbb{E}[g_i|Z_i]$ to control $\sum_{i=1}^{n}\mathbb{E}[g_i|Z_i]$, and Thm 1 to control $\sum_{l=0}^{k-1}\sum_{i=1}^{n}\big(g_i^l-g_i^{l+1}\big)$.
>
> **Q3: In Remark 4 in the fourth line, should it be ''we cannot guarantee $E[\tilde{g}_i]=0$…''?**
>
> **A**: Thank you for indicating this. It should be ``While $E_{z_i}[\tilde{g}\_i]=0$ in the i.i.d. case, we cannot guarantee $E_{z_i}[\tilde{g}\_i]=0$ in the mixing case''. We have revised it in the revised version.
>
> **Q4: It would be helpful to clarify in the text how the conditions required within the corollaries in Section 5 handle boundedness assumption of loss functions.**
>
> **A**: Thank you for indicating this issue. The applications in Section 5 require the assumption $f(\mathbf{w}_S;z)\leq M$ for any $S$ and $z$. We have added this assumption in the first paragraph of Section 5.
>
> **Q5: In the statement of Corollary 6, I think f should be stated to be convex?**
>
> **A**: Thank you for your careful reading. Yes, $f$ should be convex and we have revised it in the revision.

---

> > ### Comment · Reviewer_2VCS · 2022-11-28
> > **Authors response**
> >
> > Thank you for clarifying the points raised in my review. I will keep my initial score of 6 since the main ideas of the proof draw heavily upon those of (Bousquet et al., 2020). However, the results are decent and the paper is overall written well in my opinion.

---

### Official Review · Reviewer_PGse · 2022-10-31

**Confidence:** 3
**Correctness:** 3
**Technical Novelty And Significance:** 3
**Empirical Novelty And Significance:** Not applicable
**Recommendation:** 6

**Clarity, Quality, Novelty And Reproducibility:**

The paper is generally well-written, though it would be nice to bring out the contribution with respect to prior work better. I'm not fully sure of the novelty and originality in the techniques, maybe some comparison of the concentration bounds and proof techniques could help here.

Clarification:

In the statement of Lemma 4, the definition of $b$ seems to have something missing?

**Strength And Weaknesses:**

Strengths:

1. The paper has significant contributions to the understanding of generalization and algorithmic stability for the case of weakly-dependent data. For e.g. it obtains bounds which match the best known bounds for i.i.d. data in some cases if the dependence among the variables is weak enough.
2. The paper introduces some interesting concentration bounds for sums and Lipschitz functions of weakly dependent random variables, which could possibly have wider applications.

Weaknesses:

1. I think the message, presentation, and perhaps the bounds in the paper as well need to be distilled more to make the contribution clearer. For e.g. how do the concentration bounds in Thm 1 and 2 compare with prior work? The bounds seem interesting, but I'm not sure of the improvement here. Similarly, I'm not fully sure of the improvement in Thm 5. If the sequence is exponentially $\varphi$-mixing (or algebraically $\varphi$-mixing with a large enough degree) then the $\sqrt{n}\varphi(b)$ term should be negligible, and the improvement is mainly in the other $\sqrt{n}b \beta \Delta_n$ term. This term could potentially be significant, but I'd like to see some specific algorithmic examples where it leads to a better downstream bound. For e.g., it would be good if the authors commented on the difference between their bounds and previous work for kernel regularizations and SGD. On a related note, I'm not sure I fully understand the claim on page 2 that Mohri & Rostamizadeh (2010) get a worse than 1/sqrt(n) bound for strongly-convex problems, why should I think of $\lambda$ as not being constant (or a very small polynomial) here? Seeing some concrete examples of how the theory leads to better bounds for well-studied algorithms would make the significance of the contributions much clearer.

**Summary Of The Paper:**

The paper considers the problem of deriving generalization bounds for weakly dependent data using algorithmic stability. It builds on previous work by Mohri & Rostamizadeh (2010) which derived the first generalization bounds for this setup. This paper claim to improve on these bounds by a factor of sqrt(n). To derive these bounds, the paper develops some new moment bounds for sums and Lipschitz functions of weakly dependent random variables. Finally, the paper applies the new generalization bounds to analyze a few algorithms including kernel regularization and SGD.

**Summary Of The Review:**

Overall, the paper appears to be a good contribution to generalization theory, though I have some concerns regarding the contribution and significance.

---

> ### Author Response · Authors · 2022-11-16
> **Response to Reviewer PGse (Part I)**
>
> Thank you for your time and efforts. Below please find the responses to some specific comments.
>
> **Q1: How do the concentration bounds in Thm 1 and 2 compare with prior work?**
>
> **A**: We add a comparison of Thm 1 with prior work as follows (Remark 1).
> Under the assumption $\sum_{k=1}^{\infty}\varphi^{\frac{1}{2}}(k)<\infty$, it was shown that [1]
>
> $$
> ||\sum_{i=1}^n Z_i ||_p < C_p  \Big(  \sum_i  ||Z_i||_p^p \Big)^{ 1 / p } + C_p \Big(\sum_i ||Z_i||_2^2\Big)^{1/2},
> $$
>
> where $C_p$ is a constant depending on $p$. This bound requires an assumption involving $\sum_{k=1}^{\infty}\varphi^{\frac{1}{2}}(k)$, which is larger than $\sum_{k=1}^{n}\varphi(k)$ since $\varphi^{\frac{1}{2}}(k)\geq \varphi(k)$. For example, if $\varphi(k)=O(k^{-1})$ then $\sum_{k=1}^{n}\varphi(k)=O(\log n)$ while $\sum_{k=1}^{n}\varphi^{\frac{1}{2}}(k)=O(\sqrt{n})$. Furthermore, the bound in [1] involves $C_p$ which is not explicitly stated. As a comparison, our bound involves all explicit constants.
>
> Thm 2 is an extension of Theorem 3.1 in [2] from i.i.d. setting to $\varphi$-mixing sequences. To our knowledge, there is no prior work of this inequality for $\varphi$-mixing sequences.
>
> [1]  Xuejun, Wang, Hu Shuhe, Yang Wenzhi, and Shen Yan. On complete convergence for weighted sums of $\varphi$-mixing random variables. *Journal of Inequalities and Applications*, 1-13, 2010.
>
> [2] Bousquet, Olivier, Yegor Klochkov, and Nikita Zhivotovskiy. Sharper bounds for uniformly stable algorithms. In *Conference on Learning Theory*, pp. 610-626, 2020.
>
> **Q2: If the sequence is exponentially $\varphi$-mixing (or algebraically $\varphi$-mixing with a large enough degree) then the $\sqrt{n}\varphi(b)$ term should be negligible, and the improvement is mainly in the other $\sqrt{n}b\beta\Delta_n$ term. This term could potentially be significant. It would be good if the authors commented on the difference between their bounds and previous work for kernel regularizations and SGD.**
>
> **A**: Yes, as you indicated, the main improvement is on the $\sqrt{n}b\beta\Delta_n$ term. To clarify the significance of improving $\sqrt{n}b\beta\Delta_n$ in Mohri \& Rostamizadeh (2010) to $\beta(b+\Delta_n^2\log^2n)$ in our manuscript, let us consider kernel regularization and algebraically $\varphi$-mixing sequences, i.e., $\varphi'(k)\leq\varphi_0k^{-r}$ with $r>1$. In this case, we can balance the two terms $b\beta$ and $\varphi(b)$ to get the optimal $b\asymp \beta^{-\frac{1}{r+1}}$. The stability parameter satisfies $\beta=O(1/(n\lambda))$. In this case, the term $\sqrt{n}b\beta\Delta_n$ is of the order $\sqrt{n}\beta^{\frac{r}{r+1}}=\sqrt{n}(n\lambda)^{-\frac{r}{r+1}}$. As a comparison, the term $\beta(b+\Delta_n^2\log^2n)$ is of the order $(n\lambda)^{-\frac{r}{r+1}}$ (up to a logarithmic factor). Therefore, the improvement on $\sqrt{n}b\beta\Delta_n$ saves a factor of $\sqrt{n}$.
>
> **Q3: I'm not sure I fully understand the claim on page 2 that Mohri \& Rostamizadeh (2010) get a worse than $1/\sqrt{n}$ bound for strongly-convex problems, why should I think of $\lambda$ as not being constant (or a very small polynomial) here?**
>
> **A**: Thank you for your comment. Our aim is to get a model $w$ with a small risk $F(w)$ instead of a small
>
> $F_\lambda(w):=F(w)+\lambda\\|w\\|_K^2$.
>
> By the regularization scheme, we are essentially minimizing $F_\lambda$ (we replace $F_S$ with $F$ for simplicity). If $\lambda$ is a constant then the minimizer of $F_\lambda$ would not lead to a model $w$ with a small $F(w)$, leading to an underfitting phenomenon. One should choose a small $\lambda$ to make sure that the minimizer of $F_\lambda$ would yield a model with a small $F$. A typical choice of $\lambda\approx n^{-\alpha}$ for $\alpha>0$ is presented in Chapter 13 in [1]. Indeed, by an error decomposition, one can show
>
> $$
>   F(w_{S,\lambda}) - F(w^*) \leq  F(w_{S,\lambda}) - F_S(w_{S,\lambda}) + F_S(w^*) - F(w^*) + \lambda\\|w^*\\|_K^2.
> $$
>
>  If we assume $\\|w^*\\|_K=O(1)$ and consider an i.i.d. setting, it follows from the $O(1/(n\lambda))$-uniform stability that
>
> $$
>  F(w_{S,\lambda}) - F(w^*)=O(1/(n\lambda))+O(\lambda)+O(n^{-\frac{1}{2}}).
> $$
>
> Setting $\lambda=O(1)$ leads to a vacuous bound. The optimal choice is $\lambda=O(n^{-\frac{1}{2}})$ which is not a constant.
>
> [1] Shalev-Shwartz, Shai and Ben-David. Understanding machine learning: From theory to algorithms. *Cambridge university press*, 2014.

---

> > ### Author Response · Authors · 2022-11-16
> > **Response to Reviewer PGse (Part II)**
> >
> > **Q4: I'm not fully sure of the novelty and originality in the techniques, maybe some comparison of the concentration bounds and proof techniques could help here.**
> >
> > **A**: Thank you for the comment. Our novelty in proving Thm 1 is to introduce an indicator function to build bounded random variables for the application of a McDiarmid inequality (we have modified Remark 1 to clarify our novelty in proving Thm 1). Our novelty in proving Thm 5 is to introduce a new error decomposition to get random variables of conditional mean zero, which is required for the application of Thm 2. We have modified Remark 5 to clarify the challenge and novelty in proving Thm 5 as follows:
> >
> > The analysis in the mixing case is more challenging than that in the i.i.d. case. The analysis in Bousquet et al (2020) introduces
> >
> > $\tilde{g}\_i=E_{z_i'}E_{z_i}[f(w_{S^i};z)-f(w_{S^i};z_i)]$,
> >
> > where $z$ is independently drawn from the stationary distribution and
> > $S^i=\\{z_1,\ldots,z_{i-1},z_i',z_{i+1},\ldots,z_n\\}$.
> >
> > While $E_{z_i}[\tilde{g}\_i]=0$ in the i.i.d. case, we cannot guarantee $E_{z_i}[\tilde{g}\_i]=0$ in the mixing case due to the dependency between $z_i$ and $z_j$ ($j\neq i$). In this way, one cannot apply Thm 2 to $\tilde{g}\_i$ in the mixing case. We use a much more complicated decomposition of the generalization error in terms of $g_i$ in Eq. (4.3), which is of mean zero in the mixing case. In this process, we introduce concepts such as $S_{i,b},S_{i,b}^i$ to fully exploit the stability and mixing property. It should be mentioned that $S_{i,b},S_{i,b}^i$ have been introduced in Mohri \& Rostamizadeh (2010). However, the aim is different. These concepts are used in Mohri \& Rostamizadeh (2010) to get bounds in *expectation* for $\Phi(S):=F(w_S)-F_S(w_S)$ via a lemma in Yu (1994) for $\beta$-mixing sequence, where the concentration of $\Phi(S)$ around its expectation can be directly studied via the McDiarmid inequality for $\varphi$-mixing sequence. As a comparison, our aim is to replace the sequence $\tilde{g}\_i=E_{z_i'}E_{z_i}[f(w_{S^i};z)-f(w_{S^i};z_i)]$ (with non-zero conditional mean) by the sequence $g_i$ in Eq. (4.3) with zero conditional mean, which is then controlled by our new concentration inequality for $\varphi$-mixing sequences (Thm 2).
> >
> > **Q5: In the statement of Lemma 4, the definition of $b$ seems to have something missing?**
> >
> > **A**: Thank you for pointing out this issue. We have modified Lemma 4 in the rebuttal revision. In the rebuttal revision, we clarified that $b\in\\{0,\ldots,n\\}$ denotes the number of last points removed in $S$, i.e., $S_b=\\{z_1,\ldots,z_{n-b}\\}$.

---

### Official Review · Reviewer_Ue97 · 2022-10-31

**Confidence:** 2
**Correctness:** 3
**Technical Novelty And Significance:** 3
**Empirical Novelty And Significance:** Not applicable
**Recommendation:** 5

**Clarity, Quality, Novelty And Reproducibility:**

The new concentration bounds and stronger generalization results for stable algorithms under $psi$-mixing are novel. The techniques largely adapt prior work, but require some new ideas to do so. As discussed above, the writing clarity is the main issue with the work in its current form, and how it positions itself with respect to prior work.

**Strength And Weaknesses:**

Studying learning settings where data is non-i.i.d is a natural and important problem, especially given that the vast majority of statistical learning theory is built for the i.i.d setting while one frequently expects dependencies to occur in practice. This work is (to my knowledge), the first to show that one can recover near-optimal generalization rates that don’t depend on classical complexity measures (VC, Rademacher, etc). The main concentration inequality for $\phi$-mixing random variables also seems to be of independent interest. At a technical level, the work draws heavily from [BKZ20] and Mohri and Rostamizadeh (JMLR 2010), and the main observation seems to be that one can combine their techniques (the former giving optimal bounds in the i.i.d setting, the latter sub-optimal for $\phi$-mixing) by making a stronger assumption on the mixing process ($\psi$-mixing). Realizing this requires some effort and ties together several disparate works.

On the other hand, there are several serious issues in the presentation of the results. First and foremost, the authors claim as one of (if not the) main selling point that their generalization results improve over Mohri and Rostamizadeh by a factor of $\sqrt{n}$, and are optimal. From what I can tell, this is not true. The authors make their improvement in large part via a stronger assumption on the decay of dependence in the mixing process, so the result is incomparable to prior work in this vein. I find the introduction to be fairly misleading in this sense, and largely due to this fact do not think I can recommend the work in its current form for publication in ICLR. There are a number of other presentation issues as well. A number of crucial definitions are not properly explained (e.g. distribution of the test examples), several of the theorem/lemma statements are confusing as written (what does “Let b ∈ {0, . . . , n} denote the number of last removed in S” mean?), and the stronger dependence assumption is never motivated (how much stronger is $\psi$-mixing than $\phi$-mixing? Is this a reasonable assumption?).

Minor comments:

1. “Note $\Delta_n = O(1)$ for algebraical φ-mixing” Do you mean for $r>1$?

2. I suggest writing bounds in asymptotic notation in the intro/beginning for readability

3. It’d be nice to motivate the conditions in Thm 2 beside just being what you need for the analysis.

4. There are many other grammatical/typographical issues throughout

**Summary Of The Paper:**

The authors study generalization bounds in parametrized supervised learning when training (and test) data are not i.i.d. This setting is more realistic than the typical independence assumptions made in statistical learning theory for problems like time series (or say a Markov chain), where it is expected that the $t$-th sample depends on the prior position (observed examples). Following a long line of work on learning from non-i.i.d data, the authors consider the setting of “mixing processes,” where the dependence between samples is controlled by some mixing coefficient that decays to 0 between the $t$-th and $(t+i)$-th examples as $i$ goes to infinity. Roughly, the authors show that any uniformly stable algorithm has near optimal generalization bounds under strong assumptions on the decay of dependence across time.

In slightly greater detail, the authors consider two main settings called $\phi$-mixing and $\psi$-mixing respectively, where roughly:

$$\phi(k) = \sup_{A,B}\left|Pr(A|B) - Pr(A)\right|, \quad \quad \psi(k) = \sup_{A,B}\left|\frac{Pr(A \cap B)}{Pr(A)Pr(B)} - 1\right|,$$

Where A and B are events k time steps apart, and $\psi$-mixing is a stronger assumption than $\phi$-mixing

The authors start by proving new concentration inequalities for $phi$-mixing sequences, showing that the sum of any family of $n$ mean-0, sub-gaussian variables $\{Z_i\}$ that are functions of random variables $\{X_i\}$ drawn from a $\phi$-mixing process have bounded p-norm (matching the classical Marcinkiewicz-Zygmun inequality for the i.i.d setting up to log factors and the mixing coefficient). A similar result is shown for bounded mean, Lipshitz, coordinate-wise 0-mean variables, generalizing a result in the i.i.d setting of Bousquet, Klochkov, and Zhivotovskiy (COLT 2020).

Using these newly developed concentration inequalities, the authors generalize the optimal stability-based generalization bounds of [BKZ20] to the non-i.i.d setting (an algorithm is said to be stable if changing a single training example does not significantly shift the output value on any point). In particular, the authors show a high probability generalization bound of roughly:

$$|F(w_S)-F_S(w_S)| \leq \tilde{O}(n^{-1/2})$$

where $F$ is a bounded loss function and $w_S$ is the (parametrized) output of a uniformly stable algorithm on sample $S$, under the assumption that the data distribution is $\psi$-mixing for $\psi(k) \leq 1/k^r$ for some $r>1$ (i.e. where dependence decays at a super-linear rate). To the authors’ knowledge, this is the first such result known to achieve optimal generalization rates for non-i.i.d data.

Finally, the authors apply this result to a few popular learning paradigms, Kernel Regularization Schemes, SGD, and Feldman, Koren, and Talwar’s recent iterative localized algorithm to give generalization bounds of the form

$$|F_S(w_S)-F(w^*)| \leq \tilde{O}(n^{-1/2})$$,

Under various standard assumptions (e.g. Lipshitz, smoothness, strong convexity), where $w^*$ is the minimizer of $F$.

**Summary Of The Review:**

The authors study an interesting setting in statistical learning theory (generalization bounds under non-i.i.d data), and prove optimal bounds under strong decay of dependence across samples. On content alone the work is (in my opinion) above the bar for ICLR, but due to significant issues in presentation I cannot recommend acceptance in its current form.

---

> ### Author Response · Authors · 2022-11-16
> **Response to Ue97 (part I)**
>
> Thank you very much for your constructive comments and suggestions. We respond to your concerns one by one below.
>
> **Q1: The authors make their improvement in large part via a stronger assumption on the decay of dependence in the mixing process, so the result is incomparable to prior work in this vein.**
>
> **A**: Thank you for the comment. We agree that our generalization bounds require a stronger assumption than Mohri and Rostamizadeh (2010). Our analysis in Lemma 4 requires $\varphi'$ to relate generalization error to a sequence of random variables $g_i$ of mean conditional zero. We would also like to mention that our main techniques developed in Section 3 (Thm 1 and Thm 2) depend only on $\varphi$. Motivated by your comment, we have made many changes to make a fair comparison between our results and the existing results in Mohri and Rostamizadeh (2010) in the abstract, introduction and other sections.
>
> For example, in the **abstract** we state our contribution as follows:
>
> *In this paper, we use algorithmic stability to study the generalization performance of learning algorithms with $\psi$-mixing data, where the dependency between observations weakens over time. We show uniformly stable algorithms guarantee high-probability generalization bounds of the order $O(1/\sqrt{n})$ (within a logarithmic factor), where $n$ is the sample size.*
>
> In **Section 6**, we conclude the paper as follows:
>
> *With high probability, we develop the first stability-based generalization bounds of the order $\widetilde{O}(1/\sqrt{n})$ for learning with a mixing sequence.*
>
> We also leave the replacement of $\varphi'$ by $\varphi$ as an interesting question for further study in the conclusion.
>
> **Finally**, we would like to mention that our results only require $\varphi'$-mixing, which is weaker than $\psi$-mixing. We guess $\varphi'$-mixing would be more similar to $\varphi$-mixing than $\psi$-mixing since both $\varphi$- and $\varphi'$-coefficients measure the difference between a conditional probability and a probability (i.e., of the form $|\mathrm{Pr}(A|B)-\mathrm{Pr}(A)|$). As a comparison, $\psi$-mixing considers the difference between $1$ and the \emph{ratio} of probabilities (i.e. of the form $|1-\mathrm{Pr}(A\cap B)/\mathrm{Pr}(A)\mathrm{Pr}(B)|$). Furthermore, our main techniques in this paper (Thm 1 and Thm 2) are developed for $\varphi$-mixing sequences.
>
> **Q2: A number of crucial definitions are not properly explained (e.g. distribution of the test examples)**
>
> **A**: Thank you for your invaluable comment. Our assumption for the test example $z$ is the strongest dependent scenario, that is, the test example is assumed to follow immediately after the sample $S$ which is the same as [1]. We have added the meaning of $z$ after Eq. (4.1). As indicated in [1], a less realistic setting is that the samples are dependent but the test points are assumed to be independent of the training sample $S$.
>
> [1] Mehryar Mohri and Afshin Rostamizadeh. Stability Bounds for stationary $\varphi$-mixing and $\beta$-mixing processes. *Journal of Machine Learning Research*, 11(26):789-814, 2010.
>
> **Q3: Several of the theorem/lemma statements are confusing as written (what does ''Let $b \in\\{0, . . . , n\\}$ denote the number of last removed in $S$'' mean?)**
>
> **A**: Thank you for pointing out this issue. We have modified Lemma 4 in the rebuttal revision. In the rebuttal revision, we clarified that $b\in\\{0,\ldots,n\\}$ denotes the number of last points removed in $S$, i.e., $S_b=\\{z_1,\ldots,z_{n-b}\\}$.

---

> > ### Author Response · Authors · 2022-11-16
> > **Response to Ue97 (part II)**
> >
> > **Q4:  The stronger dependence assumption is never motivated (how much stronger is $\psi$-mixing than $\phi$-mixing? Is this a reasonable assumption?)**
> >
> > **A**: Thank you for your constructive comments. Both $\varphi$-mixing and $\psi$-mixing belong to strong mixing conditions. From the Definition 1 and Definition 2 in our paper, we can obtain that:
> > \begin{align*}
> >   \varphi(k)=\sup_{n,A\in\sigma_{n+k}^\infty,B\in\sigma_{-\infty}^n}\big|\mathrm{Pr}(A|B)-\mathrm{Pr}(A)\big|&=\sup_{n,A\in\sigma_{n+k}^\infty,B\in\sigma_{-\infty}^n}\mathrm{Pr}(A)\big|\mathrm{Pr}(A\cap B)/\mathrm{Pr}(A)\mathrm{Pr}(B)-1|\\\\
> >   &\leq \sup_{n,A\in\sigma_{n+k}^\infty,B\in\sigma_{-\infty}^n}\big|\mathrm{Pr}(A\cap B)/\mathrm{Pr}(A)\mathrm{Pr}(B)-1|\\\\
> >   &=\psi(k),
> > \end{align*}
> >  which implies that $\psi$-mixing is stronger than $\varphi$-mixing. The inequality of the numerical relationship between the two is as follows: $\varphi(k)\leq\frac{1}{2}\psi(k)$ which is presented in page 109 in [1]. As for the mixing rates, both of them can be at least exponentially fast if $\varphi(k)< \frac{1}{2}$ ([2], Theorem 4) and $\psi(k)< 1$ ([3], Lemma 8 and Theorem 5). In fact, the construction methods for some $\varphi$-mixing and $\psi$-mixing sequences are also very similar [4].
> >
> > [1] Richard C. Bradley. Basic Properties of Strong Mixing Conditions. A Survey and Some Open Questions. *Probability Surveys*, 2:107-144, 2005
> >
> > [2] Yu.A. Davydov. Mixing conditions for Markov chains. *Theory Probab. Appl.*, 18 (1973) 312-328.
> >
> > [3] J.R. Blum, D.L. Hanson, and L.H. Koopmans. On the strong law of large numbers for a class of stochastic processes. *Z. Wahrsch. verw. Gebiete*, 2(1963), 1-11.
> >
> > [4] H. Kesten and G.L. O’Brien. Examples of mixing sequences. *Duke Math.J.* 43(1976) 405-415.
> >
> > **Q5: ``Note $\Delta_n=O(1)$  for algebraically $\varphi$-mixing'' Do you mean for $r>1$ ?**
> >
> > **A**: Thank you for your careful reading and invaluable comment. We have changed it in the revised version to "Note $\Delta_n=O(1)$  for algebraically $\varphi$-mixing with $r>1$".
> >
> > **Q6: I suggest writing bounds in asymptotic notation in the intro/beginning for readability**
> >
> > **A**: Thank you for the suggestion. We have modified our bounds in asymptotic notations in the intro/beginning for readability.
> >
> > **Q7: It'd be nice to motivate the conditions in Thm 2 beside just being what you need for the analysis.**
> >
> > **A**: Thank you for your suggestion. The assumption $\big|E_{Z_{[n]\backslash[i]}}[g_i(Z)|Z_i]\big|\leq M$ is a conditional boundedness assumption which is standard for concentration inequalities.
> > The assumption $E_{Z_i}[g_i(Z)|Z_{[n]\backslash[i]}]=0$ implies that the $g_i$ is of mean zero conditioned on any fixed $Z_{[n]\backslash[i]}$, which is stronger than $E[g_i]=0$ since the conditional expectation holds for any fixed $Z_{[n]\backslash[i]}$. The Lipschitz assumption implies that $g_i$ is insensitive to the change of any single example, which implies that $g_i$ is concentrated around its expectation by McDiarmid's inequality. We have added these motivations before Thm 2.
> >
> > **Q8: There are many other grammatical/typographical issues throughout.**
> >
> > **A**: Thank you for indicating this. We have proofread the paper several times and have corrected these grammatical/typographical issues in the revision.

---

> > > ### Comment · Reviewer_Ue97 · 2022-11-19
> > > **Rebuttal Reviewer Response**
> > >
> > > We thank the authors for thoroughly addressing our concerns. The edited version of the submission fixes the presentation issues and is above the bar for ICLR.

---

> > > > ### Author Response · Authors · 2022-11-27
> > > > **Thank you**
> > > >
> > > > Thank you for your feedback on our revision. We are very glad to know that our revision has addressed the presentation issues.

---

### Official Review · Reviewer_HM4R · 2022-10-31

**Confidence:** 2
**Correctness:** 4
**Technical Novelty And Significance:** 3
**Empirical Novelty And Significance:** Not applicable
**Recommendation:** 8

**Clarity, Quality, Novelty And Reproducibility:**

I believe that the work is novel and of good quality. I point out some typos:
- Pg. 2, “recent breakthrough” -> “recent breakthroughs”
- Pg. 3, “most stability analysis implies” -> “most stability analyses imply”
- Throughout the paper: please use “mixing sequences” rather than “mixing sequence”, e.g., in pg. 3, instead of “studying learning bounds with … mixing sequence”, replace with “studying learning bounds with … mixing sequences”
- Pg. 4, the paragraph starting with “Intuitively speaking” is not clear
- In Thm. 1 and Rmk. 1, why does the dependence on p not scale as p^{½}? It seems that it should because the tail assumption is sub-Gaussian
- Thm. 2, in the second bullet point and in the conclusion, write g_i(Z) rather than g_i
- Second display eq. In Remark 2, there is an extra parenthesis
- Pg. 5, do not write z_i’(z_i’’) as it looks like z_i’ is a function of z_i’’, instead write (resp. z_i’’)
- Statement of Lem. 4 needs to be checked for grammar
- Remark 4, “one guarantee E[..]” -> “one guarantees E[..] = 0”
- Conclusion, “Mcdiarmid” -> “McDiarmid”

**Strength And Weaknesses:**

The paper makes a good contribution to learning theory, with improved results over prior literature.

The main weakness is that the paper provides no examples of mixing sequences (beyond the trivial i.i.d. case), which makes it difficult to understand the relevance of the work. Please provide such examples (if the authors did not do so out of a lack of space, then they can cut down on the elaboration of the various consequences in the algebraic and exponentially mixing cases in Section 5, since these are straightforward corollaries of the stated results).


**Summary Of The Paper:**

This paper proves stability-based generalization bounds under the assumption that the data is drawn from a mixing sequence (rather than assuming that the data is i.i.d.). The bounds improve upon previous results by a factor which is the square root of the sample size.

**Summary Of The Review:**

Although the paper does not motivate the mixing setting well with examples, overall I believe that the contribution is solid and hence I recommend acceptance.

---

> ### Author Response · Authors · 2022-11-16
> **Response to HM4R**
>
> Thank you very much for your time and efforts. Below please find the responses to some specific comments.
>
> **Q1: The main weakness is that the paper provides no examples of mixing sequences (beyond the trivial i.i.d. case).**
>
> **A**: Thank you for your constructive comment. We have added specific examples of $\varphi$-mixing and $\psi$-mixing in Section 3 in the revision (we remove some remarks in Section 5 due to space limits). There are many works on examples of mixing sequences. Corollary 2.2.7 in the paper [1] generates $\psi$-mixing Markov chains. The work [2] constructed some strictly stationary Markov
> chains that are $\varphi$-mixing in the page 213-214. The paper [3] utilizes a very similar construction method to generate $\varphi$-mixing and $\psi$-mixing sequences respectively in Thm 2 and Thm 3. We provide below specific examples of $\varphi$-mixing and $\psi$-mixing mentioned in [3], respectively.
>
> We consider the random variables $V_n$, $U_n$ and $S_n$ for $n\in \mathbb{Z}$ defined on the probability space $(\Omega, \mathcal{F},P)$, which are independent of each other and are distributed as follows: for all $n\in \mathbb{Z}$, $P(V_n=i)=\beta_i$, $i=0,1$ and $0<\beta_0<\beta_0+\beta_1=1$; $P(U_n=k)=p_k\geq 0$, $k=0,1,\cdots$ and $\sum_{k=0}^{\infty}p_k=1$; $P(S_n=j)=\gamma_j$, $j=0,1,2$ and $0<\gamma_0<\gamma_0+\gamma_1<\gamma_0+\gamma_1+\gamma_2=1$.
>
> In the following we first give the example of $\varphi$-mixing sequence.
>
> Let $\{f_k\}$ be a non-increasing sequence such that $f_1 \leq 1, f_k \rightarrow 0$ as $k \rightarrow \infty$ and  $2\log(1-f_{k+1})\geq \log(1-f_{k})+\log(1-f_{k+2})$ for $\\{k: f_k<1\\}$. For any $0<\epsilon<\frac{1}{2}$ we define $\beta_0=\epsilon$, $\beta_1=1-\epsilon$. Define $\{p_n\}$ by
> $$
> \sum_{k=0}^{n-1} p_k= \begin{cases}\left(1-f_n\right)\left(1-f_{n+1}\right)^{-1} & \text { if } \quad f_n<1 \\\\ 0 & \text { if } \quad f_n=1 .\end{cases}
> $$
> Note that $p_k=(1-f_{k+1})(1-f_{k+2})^{-1}-(1-f_k)(1-f_{k+1})^{-1}\geq 0$ for $\\{k: f_k<1\\}$ since $2\log(1-f_{k+1})\geq \log(1-f_{k})+\log(1-f_{k+2})$. Let $X_n=U_n+\frac{1}{2} V_n+\frac{1}{4} W_n$ where $W_n=V_{n-U_n}$. Then $\\{X_n, n \in \mathbb{Z}\\}$ is a $\varphi$-mixing sequence with $(1-\epsilon) f_k \leq \varphi(k) \leq f_k$.
>
> Next we provide an example of $\psi$-mixing sequence.
>
> Let $\{g_k, k \geq 1\}$ be a sequence such that $g_1-g_2=1$, $g_k \rightarrow 0$ as $k \rightarrow \infty$ and $2g_{k+1}\leq g_{k}+g_{k+2}$, $k=1,2,\cdots$. For any $\epsilon \in(0,1)$, Let $\gamma_2=\epsilon$, $\gamma_0=\gamma_1=\frac{1}{2}(1-\epsilon)$, $\beta_0=\beta_1=\frac{1}{2}$. Let $p_0=0, p_k=g_k-2g_{k+1}+g_{k+2}, k=1,2, \cdots$. Note that $p_k\geq 0$ for $k=0,1,\cdots$. Let $U_n, V_n$ and $S_n$ be as before. Let $Z_n=S_n I_{[S_n=0 \  or \ 1]} +V_{n-U_n} I_{[S_n=2]}$, where $I$ is the indicator function. Finally we define $X_n=V_n+2Z_n$. Then $\\{X_n, n \in \mathbb{Z}\\}$ is $\psi$-mixing sequence with $\epsilon(1+\epsilon)^{-1} g_k \leq \psi(k) \leq \exp \left[\epsilon(1-\epsilon)^{-1} g_k\right]-1$.
>
> [1] Longla, M., Mous-Abou, H. \& Ngongo, I.S. On Some Mixing Properties of Copula-Based Markov Chains. *Journal of Statistical Theory and Applications.* 21, 131–154 (2022).
>
> [2] M. Rosenblatt. *Markov Processes, Structure and Asymptotic Behavior.* Springer-Verlag, Berlin, 1971.
>
> [3] H. Kesten and G.L. O’Brien. Examples of mixing sequences. *Duke Math. J.* 43(1976) 405-415.
>
> **Q2: In Thm. 1 and Rmk. 1, why does the dependence on $p$ not scale as $p^{\frac{1}{2}}$? It seems that it should because the tail assumption is sub-Gaussian**
>
> **A**: Thank you for your comments. If we assume $Z_i$ is bounded, then one can show the summation of $Z_i$ is sub-Gaussian, i.e., with probability $1-\delta$ we can get a tail bound involving $\sqrt{\log(1/\delta)}$. In Thm 1, we assume $Z_i$ is sub-Gaussian, which already involves a factor of $\sqrt{\log(1/\delta)}$ to bound $Z_i$. From $Z_i$ to $\sum_{i=1}^{n}Z_i$ introduces another factor of $\sqrt{\log(1/\delta)}$. Therefore, the estimation of $\sum_{i=1}^nZ_i$ then involves two factors of $\sqrt{\log(1/\delta)}$, which shows a linear dependence on $p$ in the tail bound.
>
> **Q3: Some typos**
>
> **A**: Thank you for your very careful reading. We have addressed all these typos in the rebuttal revision.

---

### Official Review · Reviewer_enwM · 2022-11-01

**Confidence:** 4
**Correctness:** 4
**Technical Novelty And Significance:** 3
**Empirical Novelty And Significance:** Not applicable
**Recommendation:** 6

**Clarity, Quality, Novelty And Reproducibility:**

For the most part, the writing is clear. There are a couple of points of order:

- As I understand it, the paper's conclusions hold only for $\psi$-mixing processes (since the control on $\varphi'$ is derived through $\psi$). This means that the title is not representative, and should be updated to $\psi$-mixing instead of $\varphi$-mixing.
- I think it's worth it to explicitly define the expectation with subscript notation being used (in my understanding, everything not in the subscript is conditioned on, and only the subscripted variables are integrated over).

**Strength And Weaknesses:**

I think this paper is quite timely, and interesting. The extension of the recent iid stability breakthroughs to mixing processes was just waiting to happen, and the submission does a fine job of delivering on this. I think the paper also does a good job at characterising the existing work, and in treating a technically involved subject in a simple way. The proofs are, to my reading, correct, although I have not checked every detail.

There's perhaps two points of weakness to consider:

- To an extent, these methods at a high level look like an integration of [BKZ] and [MR]. While the authors briefly discuss the differences from [BKZ] in the proof technique (the evacuating $b$), this difference appears to be present already in [MR]. I think a clear discussion of the specific challenges thrown by the integration of these two lines, and of the specific novelty relative to these will strengthen the paper.

- The paper is very theoretical, which somewhat clashes with the flavour of this conference. While certainly most large ML conferences do have a space for theoretical work, ICLR tends to lie more strongly on the empirically driven side of things. It is thus worth considering how appropriate this paper is for the venue.

**Summary Of The Paper:**

This paper develops generalisation bounds for stable learning procedures trained on a $\psi$-mixing (dependent) stochastic process. The key contribution here is an extension of a recent line of work on generalisation analysis of stable methods on i.i.d. data. Much like in the i.i.d. case, the result is stability bounds that improve in $\sqrt{n}$ factors found in prior analysis of the same.

The extension follows two steps, both of which are mixing versions of the approach of Bousquet et al. [BKZ] - first, the authors derive a MacDiarmid type tail bound for sums of functions of $\varphi$-mixing processes along the lines of Theorem 4 of [BKZ]. Next this with a one-point ghost replacement trick is used to derive stability bounds along the lines of Lemma 7 of [BKZ]. The main novelty lies in this second part - to handle the dependence of the stream, the authors extend this technique to control the generalisation error in terms of the risks of a model trained with a replaced $i$th sample along with removing $2b$ samples around $i$. The value of $b$ serves as an (analytical) tradeoff (roughly speaking, between the loss of samples and the decorrelation engendered by removing samples). This part is strongly reminiscent of the work of Mohri and Rostamizadeh [MR].

The paper concludes with applications of the bounds derived to methods previously established to be stable, illustrating the utility of the bounds found.

**Summary Of The Review:**

I think this is an interesting paper, but have reservations for its appearance in this venue, and would additionally like a clearer description of how specific technical aspects of the argument interact with the prior work. On the whole, I lean towards recommending acceptance.

N.B. - I would have given a score of 7 if available as an option, but it's not. The score of 6 rather than 8 reflects the fact that I do have reservations about the venue, but I'm open to changing this upon discussion.

---

> ### Author Response · Authors · 2022-11-16
> **Response to enwM**
>
> Thank you very much for your time and efforts. Below please find the responses to some specific comments.
>
> **Q1: While the authors briefly discuss the differences from [BKZ] (the evacuating b), this difference appears to be present already in [MR]. A clear discussion of challenges thrown by the integration of these two lines, and of the novelty relative to these will strengthen the paper.**
>
> **A**: Thank you for your constructive comment. To handle the challenge due to the dependency among examples, we introduce a much more complicated error decomposition by introducing $g_i=E_{z_i'}\big[E_{z_i''}[f(w_{S_{i,b}^i};z_i'')]-f(w_{S_{i,b}^i};z_i)\big]$ to get mean-zero random variables. While this technique of removing $2b$ points has been presented in [MR], the aim is different. These concepts are used in [MR] to get bounds in expectation for $\Phi(S):=F(w_S)-F_S(w_S)$ via a lemma in [Yu 1994] for $\beta$-mixing sequence, while the concentration of $\Phi(S)$ around its expectation can be directly studied via the McDiarmid inequality for $\varphi$-mixing sequence. As a comparison, our aim is to replace the sequence $ \tilde{g}\_i = E_{z_i'} E_{z_i} [ f(w_{S^i};z) - f( w_{S^i}; z_i)] $
> (with non-zero conditional mean) by the sequence $g_i$ in Eq. (4.3) with zero conditional mean, which is then controlled by our new concentration inequality for $\varphi$-mixing sequences (Thm 2). We have added these discussions in Remark 5.
>
> **Q2: It is worth considering how appropriate this paper is for the venue.**
>
> **A**: Thank you for the comment. Algorithmic stability is a fundamental concept to develop algorithm-dependent generalization bounds, which has attracted increasing attention in the ICLR community. For example, stability-based analysis of several machine learning algorithms was published in ICLR in the last three years, e.g., [1, 2, 3]. Therefore, we think our paper could be appropriate for ICLR.
>
> [1]  Jian Li, Xuanyuan Luo, and Mingda Qiao. On generalization error bounds of noisy gradient methods for non-convex learning. *ICLR*, 2020.
>
> [2]  Yunwen Lei and Yiming Ying. Sharper generalization bounds for learning with gradient-dominated objective functions. *ICLR*, 2021.
>
> [3]  Shaojie Li and Yong Liu. High probability generalization bounds with fast rates for minimax problems. *ICLR*, 2022.
>
> **Q3: The paper's conclusions hold only for $\psi$-mixing processes (since the control on $\varphi'$ is derived through $\psi$). This means that the title is not representative, and should be updated to $\psi$-mixing instead of $\varphi$-mixing.**
>
> *A*: Thank you for your constructive comment. Motivated by your comment, we have modified our title to ``Sharper Bounds for Uniformly Stable Algorithms with Stationary mixing Processes'' (currently, we do not know how to modify the title in the openreview system and will manage to do this in the camera-ready version if the paper is accepted). Here are two reasons for this modification. First, our paper considers two mixing-processes. We develop concentration inequalities for $\varphi$-mixing processes in Section 3 (Thm 1 and Thm 2), which are the main technical tools in this paper. We develop generalization bounds for $\psi$-mixing processes. The reason we resort to $\psi$-mixing is that it has the symmetry property, while $\varphi$-mixing does not.  Second, we would like to mention that our results only require $\varphi'$-mixing, which is weaker than $\psi$-mixing. We guess $\varphi'$-mixing would be more similar to $\varphi$-mixing than $\psi$-mixing since both $\varphi$- and $\varphi'$-coefficients measure the difference between a conditional probability and a probability (i.e., of the form $|\mathrm{Pr}(A|B)-\mathrm{Pr}(A)|$). As a comparison, $\psi$-mixing considers the difference between $1$ and the *ratio of probabilities* (i.e. of the form $|1-\mathrm{Pr}(A\cap B)/\mathrm{Pr}(A)\mathrm{Pr}(B)|$). Moreover, the concept of time-reversed $\varphi$-mixing is mentioned in [1], which we may be able to make good use of to get rid of the dependence on $\psi$-mixing in the future work.
>
> We also change several statements in the comparison of our bounds with the bounds in [MR] to clarify our contributions.
>
> [1] Richard C. Bradley. Basic Properties of Strong Mixing Conditions. A Survey and Some Open Questions. *Probability Surveys*, 2:107-144, 2005.
>
> **Q4: I think it's worth it to explicitly define the expectation with subscript notation being used.**
>
> **A**: Thank you for your comment. We have replaced $\big|E[g_i(Z)|Z_i]\big|$ with $\big|E_{Z_{[n]\backslash[i]}}[g_i(Z)|Z_i]\big|$ and replaced $E[g_i(Z)|Z_{[n]\backslash[i]}]$ with $E_{Z_i}[g_i(Z)|Z_{[n]\backslash[i]}]$ for clarity in Theorem 2 in the rebuttal revision.

---

### Official Review · Reviewer_uJFn · 2022-11-03

**Confidence:** 3
**Correctness:** 4
**Technical Novelty And Significance:** 3
**Empirical Novelty And Significance:** Not applicable
**Recommendation:** 6

**Clarity, Quality, Novelty And Reproducibility:**

The paper is written clearly.
The authors give quality results and the work is indeed novel.
There are no experiments, so reproducibility is not applicable.

**Strength And Weaknesses:**

**Strengths:**
- There is good precedent for studying generalization bounds via stability for i.i.d. data:
  * Feldman--Vondrak, COLT 2019
  * Bousquet--Klochkov--Zhivotovskiy, COLT 2020
  * Klochkov--Zhivotovskiy, NeurIPS 2021
- Previous work on stability bounds for $\varphi$-mixing sequences are limited but important:
  * Mohri--Rostamizadeh, JMLR 2010
- The main contributions of this work are sufficiently technical, but also
  adaptations of different previous works for $\varphi$-mixing setting:
  * The moment bound for weakly dependent random variables (Theorem 1) extends
    [Ren--Liang, Statistics and Probability Letters 2001].
  * The concentration inequality for $\varphi$-mixing sequences (Theorem 2)
    builds on [Bousquet--Klochkov--Zhivotovskiy, COLT 2020].
  * The general uniformly-mixing stability bound (Theorem 5) is applied to
    several useful instances (e.g., SGD).

**Weaknesses:**
- Theorem 5 and all of the corollaries in Section 5 use
  $\varphi'(b)$, which in turn is a function of $\psi(k)$. The paper should
  more explicit about this --- it doesn't seem like the authors draw any
  attention to it. They do, however, note that ``$\psi$-mixing is stronger
  than $\varphi$-mixing'', so this should be reconciled since the final
  bounds are not solely a function of $\varphi(k)$.
- It would be helpful to mention soon after Definition 2 that we will use
  $\psi$-mixing to give a bound on the stability analysis version of
  $\varphi'$-mixing in Lemma 3. Otherwise, it's not clear why Definition 2 is
  relevant, and then Lemma 3 is somewhat surprising in that the bound isn't
  given in terms of $\varphi(k)$.

**Suggestions:**
- [page 2] Typo: "general bound" --> "generalization bound"
- [page 4] Clarification: In Theorem 1 we say $X_1, \dots, X_n$ are
  $\varphi$-mixing distributions, but the definition for $\varphi$-mixing is
  given in terms of an infinite sequence of random variables. These ideas could
  be better connected. We can say something similar about Theorem 2.
- [page 5] Suggestion: Discuss the distribution from which $z$ is drawn in more
  detail. You say that $z$ depends on $S$, but can you comment more about
  what kind of assumptions are reasonable or previously studied?
- [page 9] Typo: The equation in Corollary 9 should end with a comma, not a period.

**Summary Of The Paper:**

This paper uses the notion of "algorithmic stability" to study the
generalization performance of learning algorithms that train on data drawn from
a $\varphi$-mixing sequence. The authors give new high-probability guarantees
for uniformly stable algorithms that improve on previous state-of-the-art
results by a factor of $O(\sqrt{n})$ [Mohri-Rostamizadeh, JMLR 2010]. They then
apply their general mixing stability bound (Theorem 5) to a few special cases
for polynomially-bounded and exponentially-bounded $\varphi'(k)$-sequences:
kernel regularization schemes, SGD, and an iterative localization algorithm.

**Summary Of The Review:**

This is a good paper that could be accepted to ICLR. The target audience is
probably better suited for COLT since this work solely focuses on
generalization bounds for $\varphi$-mixing sequences (a method for quantifying
decay of correlation), but the results are of likely of interest to a broader
crowd.

---

> ### Author Response · Authors · 2022-11-16
> **Response to Reviewer uJFn**
>
> Thank you very much for your time and efforts. We have addressed the typo indicated by you. Below please find the responses to some specific comments.
>
> **Q1: Theorem 5 and all corollaries in Section 5 use $\varphi'(b)$, which in turn is a function of $\psi(k)$. The paper should more explicit about this --- it doesn't seem like the authors draw any attention to it.**
>
> **A**: Thank you for the suggestion. We agree and make this clear by adding this sentence before Lemma 4:
>
> *We will use $\varphi'(b)$ in Theorem 5 and all corollaries in Section 5. The underlying reason is that we need to remove $2b$ points around $z_i$ to get $S_{i,b}$ for the application of Theorem 2. An upper bound  $ |F(w_S) - E_{z_i''}  [f(w_{S_{i,b}}; z_i'' )]  | $ requires to use $\varphi'(b).$*
>
> **Q2: It would be helpful to mention soon after Definition 2 that we will use $\psi$-mixing to give a bound on the stability analysis version of $\varphi'$-mixing in Lemma 3. Otherwise, it's not clear why Definition 2 is relevant, and then Lemma 3 is somewhat surprising in that the bound isn't given in terms of $\varphi(k)$.**
>
> **A**: Thank you for your invaluable suggestion. We have added this note after Definition 2 in the rebuttal revision.
>
> **Q3: Clarification: In Theorem 1 we say $X_1,...,X_n$  are $\varphi$-mixing distributions, but the definition for $\varphi$-mixing is given in terms of an infinite sequence of random variables. These ideas could be better connected. We can say something similar about Theorem 2.**
>
> **A**: Thank you for your comment. We have made changes by saying ''Let $X_1,\ldots,X_n$ be a finite contiguous subsequence from a $\varphi$-mixing sequence'' in Theorem 1. We have also made similar changes in Theorem 2.
>
> **Q4: Suggestion: Discuss the distribution from which $z$ is drawn in more detail. You say that $z$ depends on $S$, but can you comment more about what kind of assumptions are reasonable or previously studied?**
>
> **A**: Thank you for your invaluable comment. Our assumption for the test example $z$ is the strongest dependent scenario, that is, the test example is assumed to follow immediately after the sample $S$ which is the same as [1]. We have added the meaning of $z$ after Eq. (4.1). As indicated in [1], a less realistic setting is that the samples are dependent but the test points are assumed to be independent of the training sample $S$.
>
> [1] Mehryar Mohri and Afshin Rostamizadeh. Stability Bounds for stationary $\varphi$-mixing and $\beta$-mixing processes. *Journal of Machine Learning Research*, 11(26):789-814, 2010.

---

> > ### Comment · Reviewer_uJFn · 2022-11-28
> > **Followup response**
> >
> > Thank you to the authors for their detailed responses to all of the reviewers. I am leaving my score unchanged.

---

### Decision · Program_Chairs · 2023-01-20

**Decision:**

Accept: poster

**Justification For Why Not Higher Score:**

The proofs seemed largely to be the marriage of the techniques of two previous papers.  This paper is not a strong fit with the interests of many ICLR attendees.


**Justification For Why Not Lower Score:**

Please see the above.


**Metareview: Summary, Strengths And Weaknesses:**

The authors obtain strong new bounds for learning with data that is not i.i.d., but mixing, using stability.  This is a fundamental, broadly applicable, advance on the topic of central interest, with clean, interpretable results.

**Note From Pc:**

if the above contains the word "oral" or "spotlight" please see: "oral" presentation means -> notable-top-5% and "spotlight" means -> notable-top-25%. As stated in our emails, we are disassociating presentation type from AC recommendations